# Genome-wide association meta-analysis identifies risk loci for abdominal aortic aneurysm and highlights PCSK9 as a therapeutic target

Abdominal aortic aneurysm (AAA) is a common disease with substantial heritability. In this study, we performed a genome-wide association meta-analysis from 14 discovery cohorts and uncovered 141 independent associations, including 97 previously unreported loci. A polygenic risk score derived from meta-analysis explained AAA risk beyond clinical risk factors. Genes at AAA risk loci indicate involvement of lipid metabolism, vascular development and remodeling, extracellular matrix dysregulation and inflammation as key mechanisms in AAA pathogenesis. These genes also indicate overlap between the development of AAA and other monogenic aortopathies, particularly via transforming growth factor β signaling. Motivated by the strong evidence for the role of lipid metabolism in AAA, we used Mendelian randomization to establish the central role of nonhigh-density lipoprotein cholesterol in AAA and identified the opportunity for repurposing of proprotein convertase, subtilisin/kexin-type 9 (PCSK9) inhibitors. This was supported by a study demonstrating that *PCSK9* loss of function prevented the development of AAA in a preclinical mouse model.

Abdominal aortic aneurysm (AAA) is a life-threatening condition in which progressive expansion of the infrarenal aorta may lead to rupture, which is associated with high mortality. Approximately 4% of the US population over 65 years of age is affected by AAA, resulting in ~41,000 deaths annually[1,2].

AAA is often discovered incidentally or as a result of screening programs in certain demographic groups. Current US Preventive Services Task Force guidelines recommend screening via duplex ultrasonography in men aged 65–75 years who have ever smoked[3], because men develop AAA at three to four times the rate of women[4] and smoking is a key risk factor[5]. The mainstay of management is longitudinal surveillance until the aneurysm size reaches the point at which the risk of rupture exceeds the risk of repair[6]. This disease surveillance period, which may last several years, represents an ideal opportunity to intervene and prevent disease progression. Unfortunately, there are currently no approved pharmacological therapies for the prevention and treatment of AAA. Although multiple pharmacological therapies have been previously proposed, based on compelling biology and promising evidence from preclinical model systems, including angiotensin-converting enzyme inhibitors, angiotensin receptor blockers, matrix metalloproteinase inhibitors and statins, to date, none have been shown to affect aneurysm growth or rupture in human trials[6].

Over the last two decades, large-scale genetic analyses have been instrumental in revealing new targets and promising therapies for atherosclerotic conditions[7,8]. Previous genome-wide association studies (GWAS) of AAA have revealed 24 genomic risk loci for AAA[9–13], but a substantial portion of AAA heritability remains unexplained. Here we leveraged genetic data across 17 studies to (1) perform a genetic discovery analysis for AAA with substantially higher numbers of participants with AAA than previous studies (fivefold increase); (2) create and test the predictive power of polygenic risk score (PRS) derived from this analysis; (3) prioritize causal genes and pathways leading to disease;

✉e-mail: tanmoy63@gmail.com; cristen@umich.edu; damrauer@upenn.edu

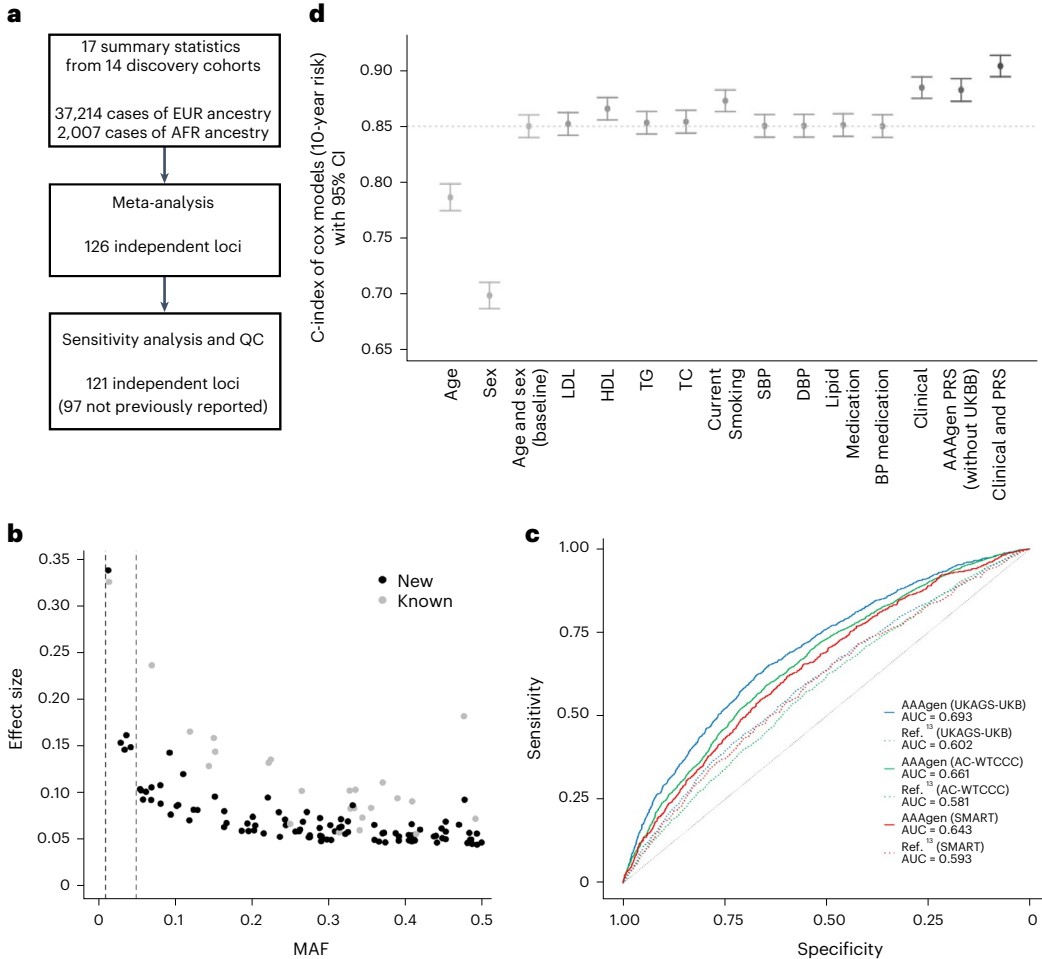

**Fig. 1 | GWAS meta-analysis and PRS of AAA. a**, Flowchart of GWAS meta-analysis. The initial analysis generated 126 genome-wide significant loci but five were excluded based on sensitivity analysis and QC. **b**, MAF is plotted against effect estimates (39,221 AAA cases and 1,086,107 controls) for genome-wide significant index variants. The robust increase in sample size compared to previous studies allowed for the identification of new disease-associated variants with smaller effect estimates. Two dashed lines represent MAF = 0.01 and 0.05. **c**,

Performance of PRS constructed based on the current meta-analysis (AAAgen) was compared with the one in ref. 13, the largest previously published GWAS of AAA. We observed improved prediction by AUC in all validation datasets. **d**, C-index of Cox models (10-year risk) with 95% CI in UKBB (838 incident AAA cases and 329,983 controls). The baseline model includes age, age² and sex. The dashed line is at the C-index value from the baseline model. All subsequent models with clinical measurements and PRS incorporate the baseline variables.

(4) explore the spectrum of phenotypic consequences associated with AAA risk variants and (5) identify potential therapeutic targets that may help prevent and treat AAA.

## Results

### GWAS meta-analysis identifies 97 new risk loci

To identify genetic variants associated with AAA, we performed a meta-analysis of 17 individual GWAS from 14 discovery cohorts in the AAAgen Consortium (Fig. 1a, Supplementary Table 1 and Supplementary Figs. 1 and 2). Our analysis comprised 39,221 individuals with AAA (37,214 of European (EUR) ancestry and 2,007 of African (AFR) ancestry). After meta-analysis, we obtained single variant association statistics for 55.8 M variants, of which 33.4 M were present in two or more GWAS and were used for downstream analyses. We identified 126 index variants associated with AAA at a genome-wide significance threshold ($P < 5 \times 10^{-8}$; Supplementary Fig. 3 and Supplementary Table 2). None of the index variants displayed significant evidence for heterogeneity (heterogeneity test; $P > 0.05/126$) of effect estimates among the contributing GWAS (Supplementary Table 2). We observed consistent effect size estimation for index variants in a comparison between meta-analysis with or without summary statistics of AFR ancestry (Supplementary

Fig. 4). Because approximately 45% of all cases were contributed by the VA Million Veteran Program (MVP) EUR analysis, we performed another meta-analysis without this cohort and tested for nominal significance ($P < 0.05$) in both datasets as sensitivity analysis. Among 126 index variants, three rare variants (minor allele frequency (MAF) < 0.01) failed to meet this threshold (Supplementary Fig. 5 and Supplementary Table 2). Additionally, for two loci we lacked confidence, as genome-wide significant rare index variants were not supported by any additional variants in these loci. These five loci were not further investigated. We performed additional sensitivity analysis for the remaining 121 loci and observed consistent effect estimates across cohorts (Supplementary Fig. 6). We also observed consistent effect estimates (Supplementary Fig. 7 and Supplementary Table 2) and $P < 0.05$ in 80/121 loci in external replication cohorts (total 5,451 cases of EUR ancestry from FinnGen and PRS validation cohorts) that were not included in the meta-analysis. Of 121 genome-wide significant loci, 97 were not previously reported (Fig. 1b). We replicated all 24 loci that were reported previously as associated with AAA[13] with $P < 5 \times 10^{-8}$. The index variants represented a wide spectrum of allele frequencies, with six being low allele frequency (MAF = 0.01–0.05) and the rest common (MAF > 0.05). As expected, by substantially (~5-fold) increasing the number of participants with AAA

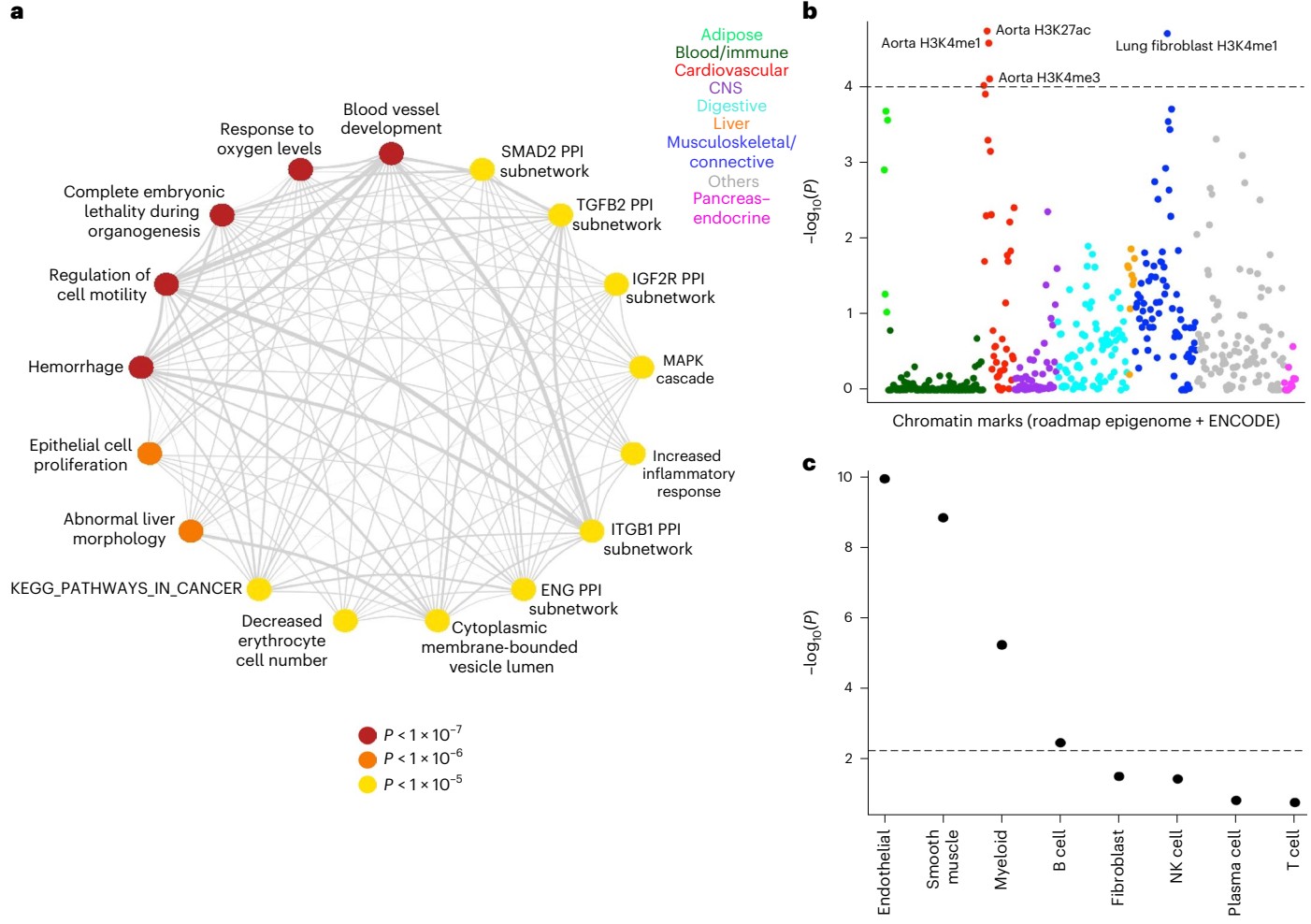

**Fig. 2 | Enrichment analysis. a**, Gene-set enrichment analysis by DEPICT. Nodes represent the representative gene sets from DEPICT (colored by *P* value). Thickness of the edges represents the overlap between gene sets. **b**, *P* values for enrichment of per-SNP heritability calculated by LDSC using tissue-specific chromatin marks. Different colors were used to classify tissues from broad categories. The dashed line represents the significance threshold after correcting for multiple testings. **c**, *P* values for estimation of the nonzero regression coefficient for each cell type calculated by RolyPoly using single-cell RNA of the aorta. The dashed line represents the significance threshold after correcting for multiple testing. NK, natural killer.

compared to previous reports[13], we were able to identify variants with lower effect estimates (Fig. 1b) that could not be discovered with smaller sample sizes. Using approximate conditional analysis, we identified 20 additional genome-wide significant variants (Supplementary Table 3) within the associated regions that were statistically independent of the 121 index variants. This resulted in 141 statistically independent associated variants.

**PRS explain AAA beyond clinical risk factors**

To evaluate the ability of our meta-analysis to explain observed disease, we generated weights for PRS using Polygenic prediction via Bayesian regression and continuous shrinkage priors (PRScs)[14]. For comparison, we also generated weights from ref. 13 (7,642 cases), the largest GWAS of AAA before this study. These two sets of weights were then used to calculate PRS in three external validation cohorts, two with control groups representative of the general population (UK Aneurysm Growth Study-UK Biobank (UKAGS-UKB) and Aneurysm Consortium-Wellcome Trust Case Control Consortium (AC-WTCCC)) and one representative of populations with cardiovascular disease (Second manifestations of ARTerial disease (SMART)). The AAAgen PRS (area under the curve (AUC); UKAGS-UKB: 0.69, AC-WTCCC: 0.66) significantly (*P* < 0.0001) outperformed that of Klarin et al.[13] PRS (AUC; UKAGS-UKB: 0.60, AC-WTCCC: 0.58) in the population-based validation

cohorts, explaining an additional 13–14% of disease-associated variance in these cohorts. AAAgen PRS performed less well in the SMART cohort, in which the controls also had prevalent (nonaneurysmal) cardiovascular disease, but still outperformed the PRS from Ref. 13 (AUC 0.64 versus 0.59, *P* < 0.0001). This also indicates the utility of PRS in distinguishing aneurysmal risk from broad cardiovascular risks for clinical applications (Fig. 1c, Supplementary Fig. 8 and Supplementary Table 4a).

To further evaluate the predictive power of the current PRS on incident AAA, we conducted analyses in data from the UK Biobank (UKBB; Fig. 1d and Supplementary Table 4b). To avoid overfitting, we performed a meta-analysis without UKBB summary statistics and generated weights for PRS. The predictive power of the model with PRS (C-index = 0.882 (0.872; 0.892)) exceeded the baseline model (C-index = 0.850, 95% confidence interval (CI) (0.840; 0.860)), as well as models with most clinical risk factors, including smoking status (C-index = 0.872 (0.863; 0.882)), and similar to the model including all tested clinical factors together (C-index = 0.884 (0.875; 0.894)). Furthermore, adding both the PRS and all clinical factors into one model yielded a C-index of 0.904 (0.894; 0.913), which represents remarkably high concordance between predicted and observed cases in a population-based cohort with notable selection bias toward healthy individuals, and substantial improvement over baseline demographics only (Δ = 0.054). This model additionally showed superiority over

**a**

Stage 1: Input—any gene within 1 mb
of 121 index variants

1. Protein-altering variation

2. Distance

3. eQTL (colocalization/TWAS)

4. PoPS

5. Monogenic disease

Stage 2: Input—523 candidate genes from stage 1

1. Expression in mouse model

2. Expression in human AAA

3. Mouse knock-out

Stage 3: Prioritize single causal gene at 121 loci
using eight indicators from stages 1 and 2

Following rules were used in order of precedence:

1. Genes with protein-altering variants (21 loci)

2. By consensus of at least 3 indicators (63 loci)

3. Nearest gene (37 loci)

**b**

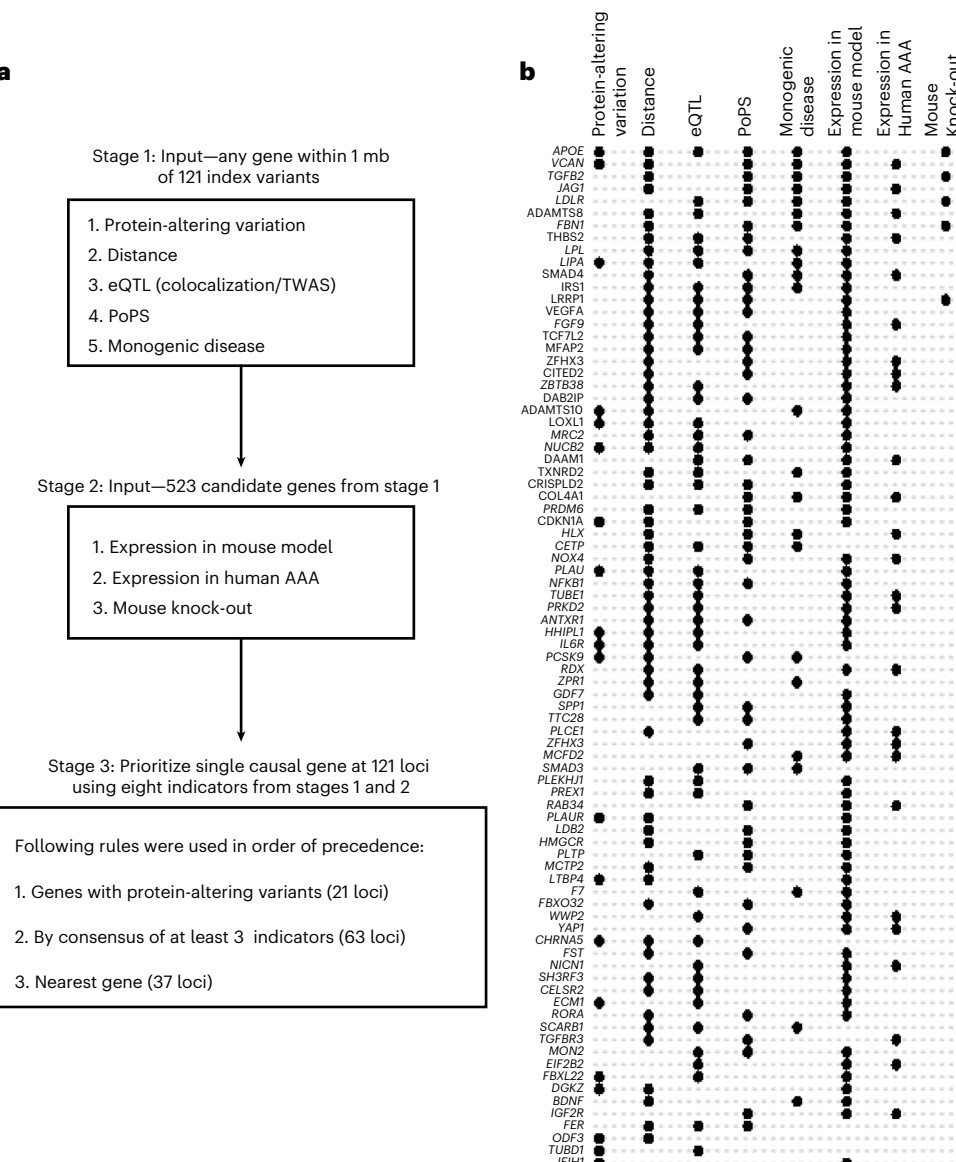

**Fig. 3 | Gene prioritization. a,** Flowchart for the gene-prioritization pipeline. The primary goal of this pipeline was to identify a single gene for each of the 121 genome-wide significant loci. Evidence of eight indicators was collected in two stages so that each prioritized gene was supported by at least one indicator from stage 1. At stage 1, we used five indicators to collect evidence for all genes within 1 Mb of index variants. This procedure identified 523 candidate genes with evidence of at least one indicator. At stage 2, evidence of three additional indicators was collected for these 523 genes. Finally, at stage 3, the above eight indicators were combined in three steps in order of precedence to identify a single prioritized gene for 121 loci. **b,** Support of various gene prioritization indicators for 84 loci where a putative causal gene could be prioritized by protein-altering variants or by consensus. Rows (gene names) represent these loci and columns represent eight supporting indicators used for the prioritization. Black dots at the row/column intersection indicate support by a particular indicator for a particular gene.

the model with clinical factors only when comparing the model fit ($P$ value from the likelihood ratio test for nested models = $1.06 \times 10^{-101}$). We did not notice any notable optimism bias in the C-indexes when bootstrapping the statistics.

### Biological functions, tissues and cell types

We performed a gene-set enrichment analysis using reconstituted gene sets in DEPICT[15]. Of 14,462 reconstituted gene sets, 114 were significant at false discovery rate (FDR) < 0.01 and $P < 10^{-5}$ (Supplementary Table 5). The representative gene sets indicated diverse biological functions such as blood vessel development, hemorrhage, abnormal liver morphology, decreased erythrocyte cell number and increased inflammatory response (Fig. 2a).

To understand the causal tissues where genes in the GWAS loci might be functional, we used a stratified linkage disequilibrium score regression (LDSC)[16] analysis using histone marks from ENCODE and Roadmap Epigenomics. This analysis identified enrichment of per-single nucleotide polymorphism (SNP) heritability in H3K27ac ($P = 1.8 \times 10^{-5}$), H3K4me1 ($P = 2.6 \times 10^{-5}$) and H3K4me3 ($P = 7.8 \times 10^{-5}$) marks in aorta (Fig. 2b). This indicates significant involvement of aortic tissue biology downstream of the observed GWAS signals.

Variants identified by GWAS often affect genes that are only active in a subset of cell types in a tissue. To further investigate the involvement of particular aortic cell types, single-cell RNA (scRNA; Supplementary Fig. 9) from aorta[17] was analyzed using RolyPoly[18]. This regression-based polygenic model uses GWAS summary statistics

and scRNA data to learn a regression coefficient that captures each cell type's influence on the variance of GWAS effect estimates. Using regression coefficients, 4 of 8 cell types were found to be enriched ($P < 0.05/8$), with endothelial ($P = 1.1 \times 10^{-10}$) and smooth muscle ($P = 1.4 \times 10^{-9}$) cells being most strongly associated with AAA (Fig. 2c). The involvement of endothelial cells likely highlights the overlap with atherosclerosis, whereas the smooth muscle involvement is consistent with the medial degeneration typical of AAA pathogenesis.

### Gene prioritization identifies putative causal genes

Using our two-stage ensemble approach (Fig. 3, Supplementary Note and Supplementary Tables 2, 6–14), we were able to prioritize a single putative causal gene at the 121 genome-wide significant loci. Most of the prioritized genes (80 of 84 genes prioritized by protein-altering variation or consensus, 28 of 37 genes prioritized by distance only) were expressed in the aorta, particularly in endothelial and smooth muscle cells, as observed in bulk and scRNA-seq of abdominal aorta (Supplementary Fig. 10 and Supplementary Note).

Next, these 121 prioritized genes, as well as an additional ten genes (Supplementary Table 3) that were the gene closest to secondary signals at GWAS loci, were used to identify enriched Gene Ontology (GO) terms (Supplementary Table 15) using Enrichr[19]. We performed a sensitivity analysis using 84 genes that were prioritized by protein-altering variation or consensus (Supplementary Table 15). Based on GO molecular function annotations, we observed enrichment of genes related to lipid metabolism (*APOE, LDLR, LPL, PCSK9, PLTP, SCARB1*), transforming growth factor beta (TGF-β) signaling (*TGFBR3, TGFB2, SMAD3, TGFB3, TGFBRAP1, LTBP4, GDF7*), cytokine activity/receptor binding (*IL6R, VEGFA, INHBA, IL1F10*) and growth factor activity/receptor binding (*JAG1, FGF9, BDNF, PDGFRA, FER*). The most substantially enriched GO cellular component term was the collagen-containing extracellular matrix (Supplementary Note).

### Transcriptome-wide association study (TWAS) indicates the direction of gene expression changes

GWAS often identify noncoding variants that are hypothesized to be associated with disease via alteration of gene-expression levels of causal genes. To further explore this possibility, we integrated results from TWAS (aortic tissue reference panel[20]) with transcriptomic analysis of aortic tissue from mouse models of AAA (Supplementary Table 11). Of 121 prioritized genes, 22 had the support of both TWAS and the mouse model of AAA. Of 22 genes supported by both, ten genes were differentially expressed in the same direction during mouse AAA development and TWAS prediction. Additionally, four genes displayed mixed direction in the mouse model, that is, direction matched with TWAS for some experimental conditions. Of these 14 genes that displayed consistency in direction, 11 (Fig. 4a) were prioritized as likely causal genes at GWAS loci by our consensus approach without distance as a tie-breaker (Supplementary Note).

We further validated the 11 genes using qPCR[21] in 97 individuals with AAA and 36 without AAA. Five of these 11 genes (Fig. 4b and Supplementary Table 16) were differentially expressed ($P < 0.05/11$). Whereas all five genes were upregulated in both the TWAS prediction and in the mouse data, four genes (*LIPA, MRC2, PRKD2* and *PLTP*) were upregulated while 1 (*MFAP2*) was downregulated, when comparing expression in aneurysmal to nonaneurysmal abdominal aortic tissue. Two of these five genes (*LIPA* and *PLTP*) likely act through lipids and atherosclerosis. Of the remaining gene products, MRC2 functions via extracellular matrix remodeling in conjunction with urokinase plasminogen activator and its receptor[22] (*PLAU* and *PLAUR*, both of which were also prioritized by the GWAS); a microfibrillar glycoprotein (*MFAP2*) known to bind to FBN1 and FBN2 (ref. 23) and a serine/threonine protein kinase (*PRKD2*) involved in regulation of cell proliferation via inhibition of HDAC7 (ref. 24), encoded by a gene also prioritized by the GWAS.

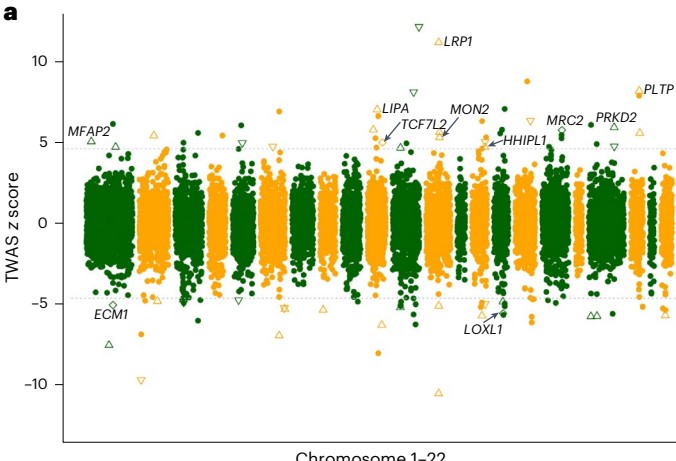

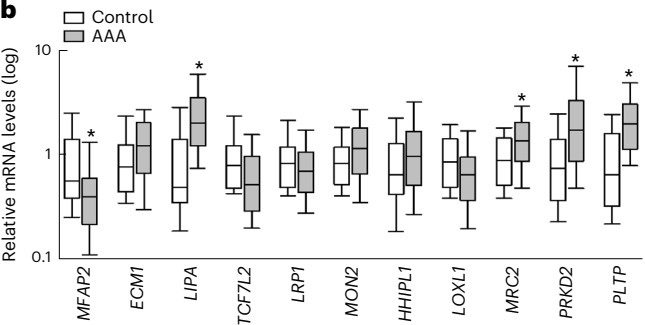

**Fig. 4 | Direction of gene-expression changes. a**, Aortic tissue-based TWAS *z* scores plotted against genomic coordinates of genes. The dashed line represents the associated *z* score for the significance threshold after multiple testing corrections. A positive or negative *z* score indicates the association of, respectively, higher or lower gene expression with AAA. Triangle/square(s) represent genes that were also differentially expressed in the mouse model of AAA. Upward/downward triangles represent genes that were observed to have high and low expression, respectively, in the mouse model. Squares represent genes where both directions were observed. Eleven prioritized genes (Results) are highlighted with text. Two colors represent odd and even number chromosomes sequentially (chromosomes 1–22). **b**, Results of qPCR in 11 prioritized genes ($n_{AAA} = 97; n_{control} = 36$). An asterisk indicates five genes with a significant (Mann–Whitney rank-sum test; two-sided *P* values < 0.05/11 after adjusting for multiple comparisons) difference in expression level between cases and controls. The boxplot is the IQR range, defined by the 25th (lower edge) and the 75th (upper edge) percentiles, with the middle bar representing the median. Whiskers define the 10th and 90th percentiles. IQR, interquartile range.

### Genetic overlap with thoracic aortic aneurysm (TAA)

Although there is human genetic evidence regarding the role of the TGF-β signaling pathway in TAA[25], which is characterized by dilation of the aortic root or the ascending/descending aorta nearest the heart itself, this GWAS provides human genetic evidence of the involvement of this pathway in AAA as well and suggests shared biology between the two diseases. To further investigate the overlap between AAA and TAA genes, we queried the association of AAA index variants in a recent TAA GWAS ($n = 1,351$)[26] and observed 24 AAA index variants associated with TAA at $P < 0.05$. These variants are generally associated with AAA with weaker effect estimates compared to TAA (Supplementary Fig. 11). This observation is consistent in two genome-wide significant common variant associations of TAA intronic to the *FBN1* (coloc; PP4: 0.94) and *TCF7L2* (coloc; PP4: 0.92; Supplementary Fig. 11) as well. Although 7 of 24 abovementioned variants (at *ECM1, DNM3, TCF7L2, CMIP, RAB34, GDF7* and *VCAN*) are also associated with lipid traits, the effect estimates

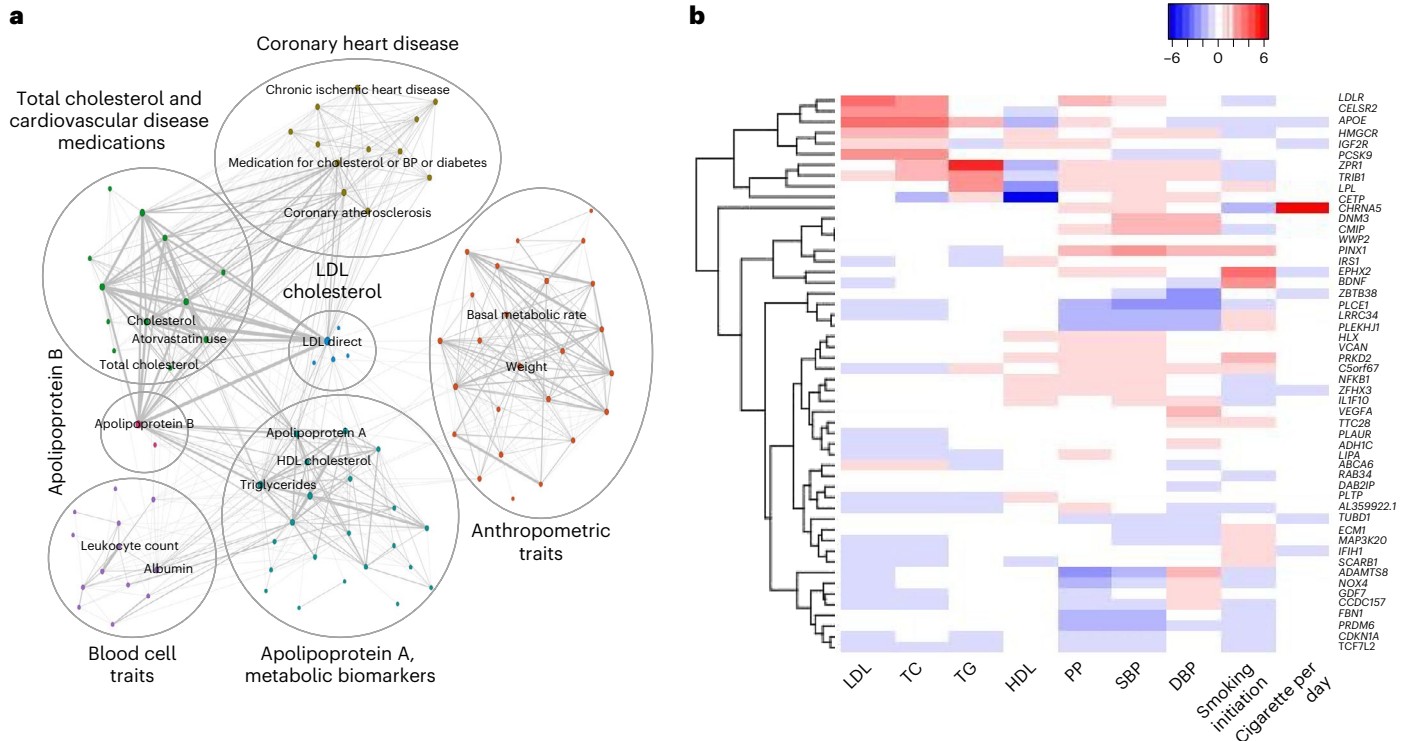

**Fig. 5 | Pleiotropy. a**, PheWAS network diagram representing seven modules (indicated by circles). **b**, Hierarchical clustering of scaled, normalized effects ($\beta$/s.e.) in established risk factor traits for 52 index variants (Results). Alleles in risk factor traits were aligned with the AAA-increasing allele. Index variants (rows) are named by the prioritized genes. PP, pulse pressure; DBP, diastolic blood pressure; SBP, systolic blood pressure.

for lipid traits are relatively weaker than either type of aneurysm for these variants (Supplementary Table 17).

Unlike AAA, ~10–20% of TAA cases have a monogenic origin[27]. Supported by varying levels of evidence, the study in ref. 28 compiled a list of 53 genes that may cause monogenic TAA because of single pathogenic variants. Many of these genes represent pathways (for example, TGF-β signaling) involved in vascular development and/or extracellular matrix organization. Among Renard's 53 genes, we observed a genome-wide significant variant for AAA within 1 Mb of 14 (*FBN1*, *ACTA2*, *SMAD3*, *SMAD4*, *SLC2A10*, *PKD2*, *TGFB2*, *HCN4*, *JAG1*, *ADAMTS10*, *TGFB3*, *COL4A1*, *VCAN* and *SMAD6*). Of note, 12 of the 14 genes were supported by at least three indicators in the gene prioritization step and nine (*FBN1*, *SMAD3*, *SMAD4*, *TGFB2*, *JAG1*, *ADAMTS10*, *TGFB3*, *COL4A1* and *VCAN*) were prioritized as the likely causal genes. Although the contribution of the abovementioned pathways is evident in both AAA and TAA, stronger effect estimates or monogenic mechanisms likely indicate a stronger contribution to TAA compared to AAA. In contrast, differences in overlap with lipid metabolism genes likely indicate larger involvement of lipid levels as a risk factor for AAA compared to TAA. Consistent with this, we observed a significant genetic correlation between lipids and AAA, but not TAA (Supplementary Fig. 12).

### Pleiotropy indicates shared heritability to lipid biology

We observed a significant genetic correlation between AAA and common cardiometabolic risk factors (Supplementary Fig. 13) as well as other vascular diseases, including coronary artery disease (Supplementary Fig. 14). To assess the pleiotropy associated with AAA risk variants and identify conditions with shared genetic risk, we performed a phenome-wide association study (PheWAS) of the 121 genome-wide significant index variants against the MRC-IEU open GWAS project[29]. Based on follow-up network analysis (Methods) using PheWAS summary statistics, we identified seven distinct modules of phenotype clusters (Fig. 5a and Supplementary

Tables 18 and 19) representing low-density lipoprotein (LDL) cholesterol, apolipoprotein B, coronary heart disease, anthropometric traits, apolipoprotein A/metabolic biomarkers, blood cell traits and total cholesterol/cardiovascular disease medications. Next, we specifically tested the association of AAA index variants with established clinical risk factors for AAA[13,30] using the largest available GWAS summary statistics of four lipids traits[31], three blood pressure traits[32] and two smoking traits[33]. We observed 52 variants (42 lipids, 11 blood pressure and 2 smoking) that were genome-wide significant ($P < 5 \times 10^{-8}$) in at least one of these traits (Fig. 5b and Supplementary Table 17). To elucidate if the same variants are associated with AAA and the risk factors, pairwise colocalization between AAA and each risk factor trait was performed. At a threshold of PP4 > 0.8, we observed colocalization of 26 AAA loci with lipids (including *PCSK9*, PP4 = 1 for both LDL and total cholesterol) and 9 loci with blood pressure (Supplementary Table 20). This set of analysis highlights that a substantial proportion of AAA loci likely function through modulating blood lipid levels, which in turn contribute to AAA development.

### Mendelian randomization (MR) identifies opportunities for lipid-modulating therapies

Given the strong contribution of lipid biology to the pathogenesis of AAA identified in our gene prioritization and PheWAS analyses, we next sought to prioritize the role of circulating lipoproteins on AAA risk. We first performed conventional inverse-variance weighted MR, finding associations between each major lipoprotein-related trait (nonhigh-density lipoprotein cholesterol (non-HDL-C), LDL cholesterol (LDL-C), HDL-C, triglycerides, ApoA1, and ApoB) and AAA (Supplementary Fig. 15, Supplementary Tables 21 and 22 and Supplementary Note). Given the substantial overlap in risk variants associated with each lipoprotein trait, we next performed an MR-Bayesian model averaging (BMA) analysis[34], an analytical tool that applies Bayesian principles to prioritize causal risk factors among correlated exposures (Supplementary

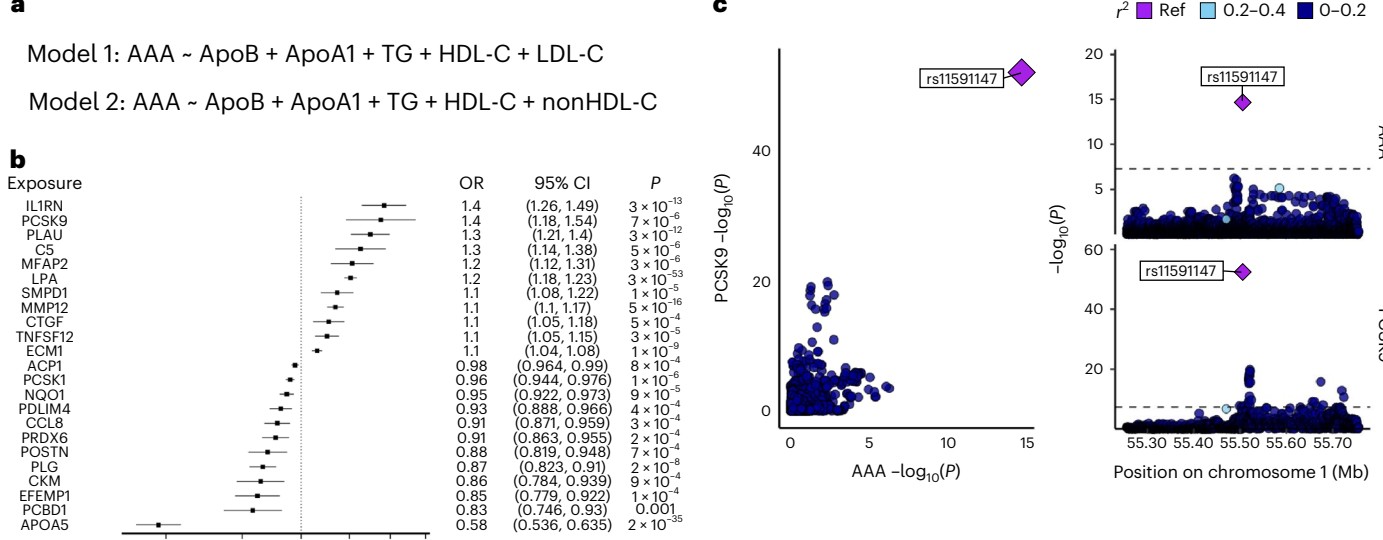

**Fig. 6 | Causal inference methods for the evaluation of AAA risk. a**, In the MR-BMA analysis to prioritize likely causal lipoprotein fractions, two separate models were used for analysis given the strong correlation between non-HDL-C and LDL-C. All other lipid fractions were included in both models. **b**, A proteome-wide MR analysis using high-confidence *cis*-acting genomic instruments for circulating plasma proteins identified 23 putative causal protein–AAA associations at two-sided Benjamini–Hochberg FDR < 0.05. ORs are depicted per unit change in protein level (1 s.d.). Genetic instruments were constructed from GWAS of circulating proteins among up to 35,559 individuals. **c**, LocusCompare plot of circulating PCSK9 protein pQTL results and AAA GWAS results demonstrating evidence of colocalization. TG, triglyceride lipoprotein.

Table 23). LDL-C and non-HDL-C emerged as the most highly prioritized causal lipoproteins for AAA risk (LDL-C marginal inclusion probability = 0.97, $P = 0.0004$; non-HDL-C marginal inclusion probability = 0.99, $P = 6 \times 10^{-5}$; Fig. 6a and Supplementary Table 24a). When comparing models containing LDL-C versus non-HDL-C variants, $r^2$ and Bayesian information criterion values demonstrate better model fit and variance explained for non-HDL-C. This finding suggests that the additional remnant cholesterol contained within the non-HDL-C subfraction is contributing to AAA risk beyond the LDL-C component (Supplementary Table 24b). This observation is consistent with prior evidence in atherosclerotic coronary artery disease[35]. There was reduced evidence that ApoA1-containing particles (ApoA1 or HDL-C) also potentially contributed to AAA (marginal inclusion probability = 0.66/0.85, $P = 0.01/0.004$).

We next sought to leverage our genetic data to screen for possible new therapeutic targets for AAA. Leveraging MR to test the association of genetically predicted circulating protein levels on genetic liability to AAA, we identified 23 putatively causal protein–AAA association (Fig. 6b,c and Supplementary Table 25). Notably, we observed that higher genetically predicted circulating PCSK9 and lipoprotein(a) were associated with increased AAA risk; this was supported by evidence of significant colocalization between PCSK9 protein quantitative trait loci (pQTL) and AAA GWAS at the *PCSK9* locus (PP = 1). In addition, higher genetically predicted circulating apolipoprotein A5 (*APOA5*) was associated with decreased AAA risk. Hyperlipidemic mice overexpressing *Apoa5* have been shown to have markedly decreased circulating remnant lipoprotein particles[36,37], and individuals with rare protein-altering variants in *APOA5* were observed to have increased levels of remnant cholesterol[38], providing further evidence of the effects of remnant cholesterol on AAA risk. These findings further support the role of lipid biology in AAA pathogenesis and the concept of lipid manipulation to prevent and treat AAA.

### PCSK9 inhibition and AAA risk in a mouse model

PCSK9 inhibition presents an attractive therapeutic for AAA given the potency of its LDL-C reduction and favorable side-effect profile[39]. Using a porcine pancreatic elastase (PPE) infusion model of AAA in C57BL/6J

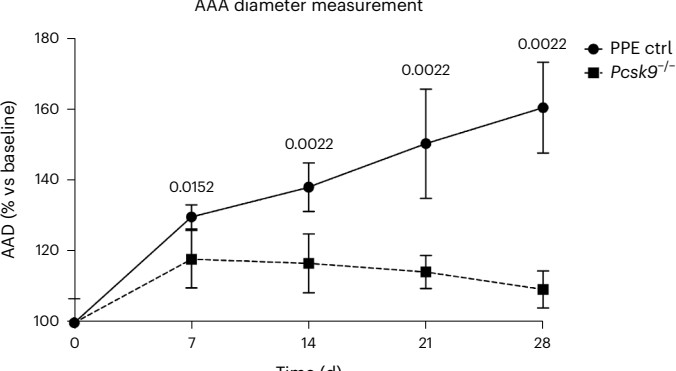

**Fig. 7 | Loss of Pcsk9 inhibits growth of experimental AAA in mouse models.** PPE surgery was performed on day 0 and maximal aortic diameter was monitored by B-mode ultrasound. Knockout of *Pcsk9* resulted in blunted AAA growth starting at experimental day 7 (PPE ctrl group n = 6; *Pcsk9*⁻/⁻ group n = 6). Graphs show aortic aneurysm diameter percentage increase versus baseline (means ± s.e.m.). Two-sided *P* values from the Mann–Whitney test were reported.

mice (Fig. 7), we investigated whether *Pcsk9* null mice demonstrated attenuated aneurysm growth compared to their wild-type counterparts. Ten-week-old mice received aortic PPE infusion as previously described[40]. We observed a significant decrease in expansion of the abdominal aortic diameter (AAD) from day 10 until day 28 for *Pcsk9* null mice compared with wild-type mice with brightness modulation (B-mode) ultrasound imaging performed 7, 14, 21 and 28 d after PPE infusion (Fig. 7; Mann–Whitney test, $P < 0.05$ on day 7; $P < 0.01$ on day 28 for *Pcsk9*⁻/⁻ versus wild type). As with previous studies[41–43], *Pcsk9*⁻/⁻ mice had lower plasma cholesterol, LDL and HDL levels compared to wild-type C57BL/6J mice (Supplementary Table 26). These experiments in *Pcsk9* null mice provide orthogonal support to our human genetics findings highlighting the role of PCSK9 in AAA development.

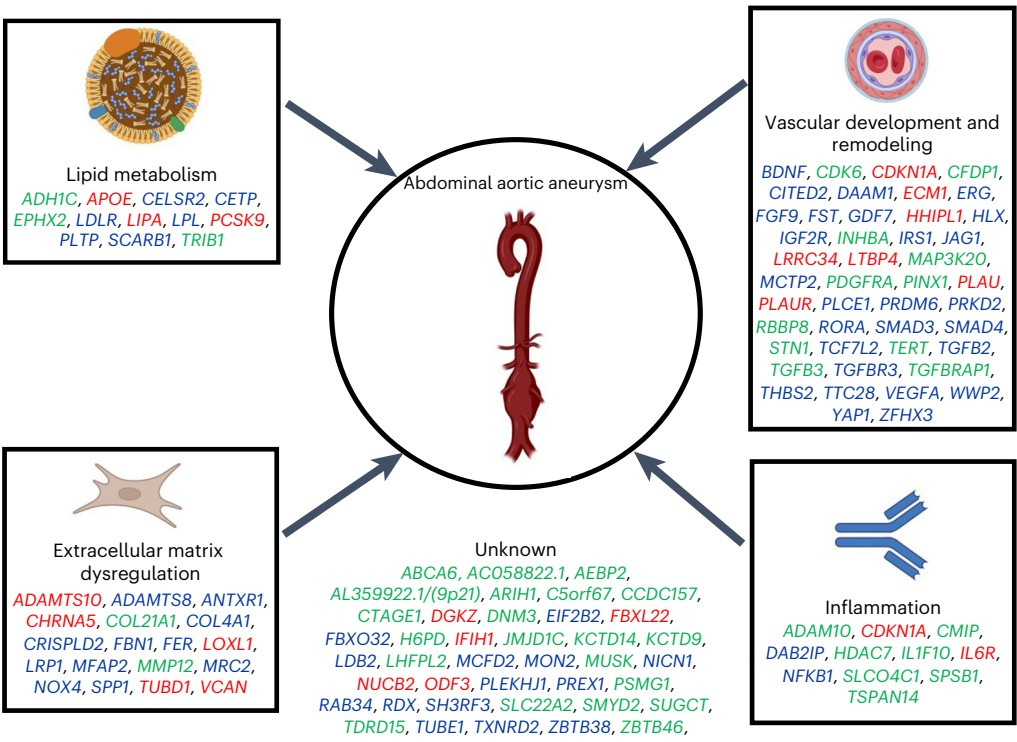

**Fig. 8 | Biological mechanisms underlying genetic loci associated with AAA.** The AAA GWAS loci for which we have identified a candidate causal gene are depicted along with the plausible relationship to its underlying biological mechanism. Loci names are based on the candidate causal gene identified in our analysis. However, the biological pathway(s) remain unclear for many associated loci and, as such, the resultant annotation may prove incorrect in some cases. Gene names are colored according to the gene prioritization scheme used to obtain the candidate (red, protein-altering variants; blue, consensus of indicators and green, nearest gene).

## Discussion

We leveraged clinical and genetic data from 14 cohorts to identify and investigate the genetic determinants of AAA in 39,221 individuals with AAA and 1,086,107 controls. We identified 141 independent AAA-associated variants in 121 loci. We confirmed 24 previously identified AAA genetic risk loci and uncovered 97 new loci. The increased power in the meta-analysis, compared to the previously published studies, was also evident in the improved performance of PRS derived from the new summary statistics. These data enabled us to prioritize putative causal genes and pathways for AAA through a combination of functional annotation and gene-expression analyses in patients with AAA and mouse AAA models. We examined the spectrum of phenotypic consequences for AAA risk variants, revealing possible mechanisms through which these variants may lead to disease. Finally, through a combination of colocalization experiments, as well as analysis in human plasma and *Pcsk9*<sup>−/−</sup> mouse AAA models, we prioritized possible therapeutic targets for the treatment and prevention of AAA.

These findings permit several conclusions. First, this research identifies important programs of the human AAA pathobiology. Our findings highlight not only lipid metabolism but also vascular development and remodeling, extracellular matrix dysregulation and inflammation as key mechanisms in the pathogenesis of AAA (Fig. 8). Although the dysregulation of these pathways in this disease has been amply demonstrated in published studies of mouse models and in diseased human tissues, their identification in the context of population-scale GWAS suggests a likely role in upstream causation. The putative causal genes identified in this research are in some cases unique to AAA, and in others, already known or suspected to have a role in other cardiovascular diseases. Drug development pipelines and clinical trials are long, expensive and complex, so new putative interventions need to

be carefully chosen. Our findings suggest that treatments that are beneficial for traits that we have now proven to be related to AAA genetically (for example, atherosclerotic vascular disease and inflammatory conditions) should also be tested for their effect on AAA.

Second, the genetic results highlight the critical role of lipids and lipid metabolism in AAA pathogenesis. Previous work has demonstrated the likely causal relationship between lipids and AAA[30,44], and current therapeutic strategies largely focus on LDL-C lowering for broad atherosclerotic risk factor modification in patients with AAA rather than specifically for disease prevention or treatment[45,46] in individuals or families at future risk of AAA. Here we expand upon these findings by observing that 42 of the 121 AAA risk loci are also associated with lipids, supporting the notion of AAA as an end-organ manifestation of atherosclerosis and that lipid-modulating therapies may have a role in the management of AAA. We additionally prioritize lipid subfractions beyond LDL-C, namely remnant cholesterol, as likely causally related to AAA. Molecules targeting the lipoprotein lipase pathway via *APOC3* (ref. [47]) are emerging therapies for cardiovascular disease but remain unstudied in patients with aneurysm. More broadly, our findings suggest that AAA shares important pathobiology with other forms of atherosclerotic vascular diseases and should be considered on this disease spectrum.

Finally, our results lend human genetic support to PCSK9 inhibition as a potential therapeutic strategy for AAA. Although current data on the effects of statins on aortic expansion are conflicted[45,48,49], human genetic evidence has overwhelmingly suggested that LDL-C reduction is likely to reduce AAA risk. PCSK9 inhibitors have been shown to dramatically reduce LDL-C [39] to levels well below those seen with other lipid lowering therapies.[39] In addition, our *Pcsk9* null mouse model demonstrated reduced AAA growth following elastase infusion in the

absence of a hyperlipidemic background, although small statistical differences in lipid fractions were observed between mice strata. Taken together, these data suggest that the relationship between circulating PCSK9 protein and AAA warrants further investigation.

Our study should be interpreted within the context of its limitations. While this stands as the largest genetic analysis to date for AAA, the demographics of the cases in our study were overwhelmingly male and EUR ancestry, thus our ability to detect sex-specific or ancestry-specific genetic associations was limited. Second, our AAA phenotype for many of the cohorts is based on electronic health record data that may have resulted in some misclassification of case status. Such misclassification should, however, reduce statistical power for discovery and, on average, bias results toward the null. Third, for a small number of loci our strategy of gene prioritization identified genes that are not well supported by literature. For example, *CELSR2* or *PLTP* instead of *SORT1* or *MMP9*, respectively. We acknowledge that our gene prioritization scheme is imperfect, as are all current methods for gene prioritization. Fourth, MR methods examining the effects of *PCSK9* inhibition on AAA reflect lifelong exposures to reduced LDL-C levels and may not represent the more acute lipid changes associated with drug administration later in life. Finally, although our analysis suggests that LDL-C reduction through *PCSK9* inhibition is likely to reduce AAA risk, it is unclear whether this treatment is likely to mitigate the progression of the disease once diagnosed. Although the results from our mouse model suggest this to be the case, further investigation into the effects of *PCSK9* inhibition on AAA growth and rupture is warranted.

In summary, we identified 141 independent variants associated with AAA risk, prioritized candidate functional genes at these loci that implicated biological pathways, developed an improved and effective PRS, explored the phenotypic consequences of AAA risk variants through PheWAS, identified candidate causal AAA genes and multiple lipid pathways and genes that may be targeted for AAA risk reduction, including *PCSK9*. These results are demonstrative of how large-scale analyses of human genetic variation coupled with clinical data can be leveraged and used for the treatment of understudied diseases.

## Online content

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

Tanmoy Roychowdhury [1,2,70] ✉, Derek Klarin [3,4,70], Michael G. Levin [5,70], Joshua M. Spin [4,6,7], Yae Hyun Rhee[4,6,7], Alicia Deng[4,6,7], Colwyn A. Headley [4,6,7], Noah L. Tsao [8], Corry Gellatly [9], Verena Zuber [10,11,12], Fred Shen[13], Whitney E. Hornsby[1], Ina Holst Laursen [14], Shefali S. Verma [15], Adam E. Locke [16], Gudmundur Einarsson [17], Gudmar Thorleifsson [17], Sarah E. Graham [1], Ozan Dikilitas [18,19,20], Jack W. Pattee[21], Renae L. Judy[8], Ferran Pauls-Verges[22], Jonas B. Nielsen[1,23], Brooke N. Wolford [24,25], Ben M. Brumpton [23,25,26], Jaume Dilmé [27], Olga Peypoch [22,27], Laura Calsina Juscafresa [28], Todd L. Edwards[29], Dadong Li [16], Karina Banasik [30], Søren Brunak [30], Rikke L. Jacobsen [14], Minerva T. Garcia-Barrio[1], Jifeng Zhang[1], Lars M. Rasmussen[31], Regent Lee[32], Ashok Handa[32], Anders Wanhainen[33,34], Kevin Mani[33], Jes S. Lindholt[35], Lasse M. Obel [35], Ewa Strauss [36,37], Grzegorz Oszkinis[37,38], Christopher P. Nelson[9], Katie L. Saxby [9], Joost A. van Herwaarden [39], Sander W. van der Laan [40], Jessica van Setten[41], Mercedes Camacho [22], Frank M. Davis[42,43], Rachael Wasikowski[44], Lam C. Tsoi[44], Johann E. Gudjonsson[44], Jonathan L. Eliason[45], Dawn M. Coleman[45], Peter K. Henke [45], Santhi K. Ganesh [1,46], Y. Eugene Chen[1], Weihua Guan[47], James S. Pankow [48], Nathan Pankratz[49], Ole B. Pedersen [50,51], Christian Erikstrup [52], Weihong Tang[48], Kristian Hveem[23,25,53], Daniel Gudbjartsson [17,54], Solveig Gretarsdottir [17], Unnur Thorsteinsdottir[17,55], Hilma Holm [17], Kari Stefansson [17,55], Manuel A. Ferreira [16], Aris Baras [16], Iftikhar J. Kullo [19], Marylyn D. Ritchie [56], Alex H. Christensen[51,57,58], Kasper K. Iversen[51,58], Nikolaj Eldrup[51,59], Henrik Sillesen[51], Sisse R. Ostrowski [14,51], Henning Bundgaard[51,57], Henrik Ullum[60], Stephen Burgess [61,62], Dipender Gill [10,63], Katherine Gallagher[42,43], Maria Sabater-Lleal [22,64], DiscovEHR*,

# Article

**Regeneron Genetics Center\*, UK Aneurysm Growth Study\*, DBDS Genomic Consortium\*, VA Million Veteran Program\*, Ida Surakka**[1,71]**, Gregory T. Jones** [65,71]**, Matthew J. Bown** [9,71]**, Philip S. Tsao** [4,6,7,71]**, Cristen J. Willer** [1,24,46,71] ✉ **& Scott M. Damrauer** [66,67,68,71] ✉

[1]Department of Internal Medicine, Division of Cardiology, University of Michigan, Ann Arbor, MI, USA. [2]Department of Genetics, Yale School of Medicine, New Haven, CT, USA. [3]Department of Surgery, Stanford University School of Medicine, Stanford, CA, USA. [4]VA Palo Alto Healthcare System, Palo Alto, CA, USA. [5]Division of Cardiovascular Medicine, Department of Medicine, University of Pennsylvania, Perelman School of Medicine, Philadelphia, PA, USA. [6]Department of Medicine, Stanford University School of Medicine, Stanford, CA, USA. [7]Stanford Cardiovascular Institute, Stanford University School of Medicine, Stanford, CA, USA. [8]Department of Surgery, University of Pennsylvania, Perelman School of Medicine, Philadelphia, PA, USA. [9]Department of Cardiovascular Sciences and NIHR Leicester Biomedical Research Centre, University of Leicester, Leicester, UK. [10]Department of Epidemiology and Biostatistics, School of Public Health, Imperial College London, London, UK. [11]MRC Centre for Environment and Health, School of Public Health, Imperial College London, London, UK. [12]UK Dementia Research Institute at Imperial College, Imperial College London, London, UK. [13]University of Michigan Medical Scientist Training Program, University of Michigan, Ann Arbor, MI, USA. [14]Department of Clinical Immunology, Copenhagen University Hospital— Rigshospitalet, Copenhagen, Denmark. [15]Department of Pathology and Laboratory Medicine, Perelman School of Medicine, Philadelphia, PA, USA. [16]Regeneron Genetics Center, LLC, Tarrytown, NY, USA. [17]deCODE genetics/Amgen Inc., Reykjavik, Iceland. [18]Department of Internal Medicine, Mayo Clinic Rochester, Rochester, MN, USA. [19]Department of Cardiovascular Medicine and the Gonda Vascular Center, Mayo Clinic Rochester, Rochester, MN, USA. [20]Mayo Clinician Investigator Training Program, Mayo Clinic Rochester, Rochester, MN, USA. [21]University of Minnesota, Minneapolis, MN, USA. [22]Unit of Genomics of Complex Diseases, Sant Pau Biomedical Research Institute (IIB Sant Pau), Barcelona, Spain. [23]HUNT Research Centre, Department of Public Health and Nursing, NTNU, Norwegian University of Science and Technology, Levanger, Norway. [24]Department of Computational Medicine and Bioinformatics, University of Michigan, Ann Arbor, MI, USA. [25]K.G. Jebsen Center for Genetic Epidemiology, Department of Public Health and Nursing, NTNU, Norwegian University of Science and Technology, Trondheim, Norway. [26]Clinic of Medicine, St. Olavs Hospital, Trondheim University Hospital, Trondheim, Norway. [27]Department of Vascular and Endovascular Surgery, Hospital de la Santa Creu i Sant Pau, Universitat Autonoma de Barcelona, Barcelona, Spain. [28]Vascular Surgery Department, Parc de Salut Mar-Hospital del Mar, Barcelona, Spain. [29]Division of Epidemiology, Department of Medicine, Vanderbilt Genetics Institute, Vanderbilt University Medical Center, Nashville, TN, USA. [30]Novo Nordisk Foundation Center for Protein Research, Faculty of Health and Medical Sciences, University of Copenhagen, Copenhagen, Denmark. [31]Department of Clinical Biochemistry, Odense University Hospital, Elite Research Centre of Individualized Medicine in Arterial Disease (CIMA), Odense, Denmark. [32]Nuffield Department of Surgical Sciences, University of Oxford, Oxford, UK. [33]Department of Surgical Sciences, Vascular Surgery, Uppsala University, Uppsala, Sweden. [34]Department of Surgical and Perioperative Sciences, Surgery, Umeå University, Umeå, Sweden. [35]Department of Cardiothoracic and Vascular Surgery, Odense University Hospital, Elite Research Centre of Individualized Medicine in Arterial Disease (CIMA), Odense, Denmark. [36]Institute of Human Genetics, Polish Academy of Sciences, Poznan, Poland. [37]Department of General and Vascular Surgery, Poznan University of Medical Sciences, Poznan, Poland. [38]Department of Vascular and General Surgery, Institute of Medical Sciences, University of Opole, Opole, Poland. [39]Department of Vascular Surgery, University Medical Center Utrecht, Utrecht University, Utrecht, The Netherlands. [40]Central Diagnostics Laboratory, Division Laboratories, Pharmacy, and Biomedical genetics, University Medical Center Utrecht, Utrecht University, Utrecht, The Netherlands. [41]Department of Cardiology, Division Heart and Lungs, University Medical Center Utrecht, Utrecht University, Utrecht, The Netherlands. [42]Department of Surgery, University of Michigan, Ann Arbor, MI, USA. [43]Department of Microbiology and Immunology, University of Michigan, Ann Arbor, MI, USA. [44]Department of Dermatology, University of Michigan Medical School, Ann Arbor, MI, USA. [45]Department of Surgery, Section of Vascular Surgery, University of Michigan, Ann Arbor, MI, USA. [46]Department of Human Genetics, University of Michigan, Ann Arbor, MI, USA. [47]Division of Biostatistics, School of Public Health, University of Minnesota, Minneapolis, MN, USA. [48]Division of Epidemiology and Community Health, School of Public Health, University of Minnesota, Minneapolis, MN, USA. [49]Department of Laboratory Medicine and Pathology, School of Medicine, University of Minnesota, Minneapolis, MN, USA. [50]Department of Clinical Immunology, Zealand University Hospital—Køge, Køge, Denmark. [51]Department of Clinical Medicine, Faculty of Health and Medical Sciences, University of Copenhagen, Copenhagen, Denmark. [52]Department of Clinical Immunology, Aarhus University Hospital, Aarhus, Denmark. [53]Department of Medicine, Levanger Hospital, Nord-Trøndelag Hospital Trust, Levanger, Norway. [54]School of Engineering and Natural Sciences, University of Iceland, Reykjavik, Iceland. [55]Faculty of Medicine, University of Iceland, Reykjavik, Iceland. [56]Department of Genetics, University of Pennsylvania, Perelman School of Medicine, Philadelphia, PA, USA. [57]Department of Cardiology, Copenhagen University Hospital–Rigshospitalet, Copenhagen, Denmark. [58]Department of Cardiology, Herlev-Gentofte Hospital, Copenhagen University Hospital, Copenhagen, Denmark. [59]Department of Vascular Surgery, Copenhagen University Hospital—Rigshospitalet, Copenhagen, Denmark. [60]Statens Serum Institut, Copenhagen, Denmark. [61]MRC Biostatistics Unit, School of Clinical Medicine, University of Cambridge, Cambridge, UK. [62]Cardiovascular Epidemiology Unit, Department of Public Health and Primary Care, University of Cambridge, Cambridge, UK. [63]Chief Scientific Advisor Office, Research and Early Development, Novo Nordisk, Copenhagen, Denmark. [64]Cardiovascular Medicine Unit, Department of Medicine, Karolinska Institutet, Center for Molecular Medicine, Stockholm, Sweden. [65]Department of Surgical Sciences, Dunedin School of Medicine, University of Otago, Dunedin, New Zealand. [66]Department of Surgery, Corporal Michael Crescenz VA Medical Center, Philadelphia, PA, USA. [67]Department of Surgery, Perelman School of Medicine at the University of Pennsylvania, Philadelphia, PA, USA. [68]Department of Genetics, Perelman School of Medicine at the University of Pennsylvania, Philadelphia, PA, USA. [70]These authors contributed equally: Tanmoy Roychowdhury, Derek Klarin, Michael G. Levin. [71]These authors jointly supervised this work: Ida Surakka, Gregory T. Jones, Matthew J. Bown, Philip S. Tsao, Cristen J. Willer, Scott M. Damrauer. \*List of authors and their affiliations appears at the end of the paper. ✉e-mail: tanmoy63@gmail.com; cristen@umich.edu; damrauer@upenn.edu

## DiscovEHR

**Dadong Li**[16] **& Aris Baras**[16]

Full list of members appears in the Supplementary Information.

Article

## Regeneron Genetics Center

Adam E. Locke[16], Jonas B. Nielsen[1,23], Dadong Li[16], Manuel A. Ferreira[16] & Aris Baras[16]

Full list of members appears in the Supplementary Information.

## UK Aneurysm Growth Study

Frank Dudbridge[69] & Nilesh J. Samani[9]

[69]Department of Population Health Sciences, University of Leicester, Leicester, UK. Full list of members appears in the Supplementary Information.

## DBDS Genomic Consortium

Karina Banasik[30], Søren Brunak[30], Ole B. Pedersen[50,51], Christian Erikstrup[52], Daniel Gudbjartsson[17,54], Unnur Thorsteinsdottir[17,55], Kari Stefansson[17,55], Sisse R. Ostrowski[14,51], Henning Bundgaard[51,57] & Henrik Ullum[60]

Full list of members appears in the Supplementary Information.

## VA Million Veteran Program

Philip S. Tsao[4,6,7,71]

Full list of members appears in the Supplementary Information.

## Methods

### Discovery cohorts

We performed a meta-analysis of 17 individual GWAS from 14 discovery cohorts in the AAAgen Consortium (Supplementary Table 1). These 14 discovery cohorts are as follows: (1) Atherosclerosis Risk in Communities (ARIC); (2) Copenhagen Hospital Biobank−Cardiovascular Disease Cohort and Danish Blood Donor Study (CHB-CVDC+DBDS); (3) Cardiovascular Health Improvement Project and Michigan Genomics Initiative (CHIP + MGI); (4) deCODE; (5) DiscovEHR; (6) Electronic Medical Records and Genomics (eMERGE); (7) Trøndelag Health study (HUNT), (8) Mayo Vascular Disease Biorepository (Mayo VDB); (9) MVP; (10) NZ AAA Genetics Study; (11) Penn Medicine biobank (PMBB); (12) Triple A Barcelona Study (TABS); (13) UKAGS and Viborg Vascular (UKAGS+VIVA); (14) UKBB (Supplementary Fig. 16). All research participants provided informed consent and local IRB approval was obtained. See Supplementary Note for a description of discovery cohorts, recruitment, case−control selection, genotyping, imputation and GWAS methods. A central level quality control (QC) was performed on summary statistics from discovery cohorts. QQ plots (Supplementary Fig. 1) and genomic control (GC) lambda were calculated for each cohort (Supplementary Table 1). Cohorts that reported variants in hg38 version were liftover to hg19 version. Cohorts that reported odds ratios (ORs) were converted to β/effect estimates using log(OR). We checked the consistency of effect estimates in nine index variants that were reported in a study discussed in ref. 12. Although we observed some heterogeneity, the effect estimates were generally consistent among well-powered studies (Supplementary Fig. 2). Variants with minor allele count of 3 and imputation $r^2 < 0.3$ were excluded from individual summary statistics.

### Meta-analysis and sensitivity analyses

Meta-analysis of 17 discovery cohort summary statistics was performed by METAL[50] in s.e. mode with GC correction. After meta-analysis, variants that were present in only one cohort were excluded from downstream analysis. For sensitivity analyses, meta-analysis summary statistics were generated in the same approach, leaving one cohort out.

### Definition of loci

Independent loci were defined as variants >1 Mb and >0.25 cM apart with at least one genetic variant associated with AAA at a genome-wide significance threshold of $P < 5 \times 10^{-8}$. Index variants are the variants with the lowest association $P$ value in every locus. Index variants were tested for independence using GCTA COJO[51] (cojo-joint) as described below.

### External replication cohorts

We used three external cohorts for the replication of 121 loci identified in the meta-analysis. We obtained publicly available AAA summary statistics (2,434 cases and 288,638 controls) from Finngen (https://r7.finngen.fi/pheno/I9_ABAORTANEUR). Additionally, we used two PRS validation sets (Supplementary Note) for this purpose. In loci where index variants were not present in external cohorts, we used variants that are in linkage disequilibrium (LD; >0.8) with index variants. For 103 loci, we could find index/LD variant in >1 cohort. We performed a meta-analysis for these variants using METAL[50] in s.e. mode. For 18 loci, we reported association statistics from Finngen only.

### Conditional analysis

To perform conditional analysis, we first defined the loci as ±1 Mb from each of the 121 index SNPs. GCTA COJO[51] and specifically the cojo-cond function was performed iteratively using a reference panel created from individuals in PMBB that represented the demographics of AAAgen (17:1 EUR to AFR ancestry). Iterations were performed at each locus until a minimum number of independently, genome-wide significant SNPs were identified. The maximum number of iterations performed was 4 at the rs10455872 locus. Next, we performed a joint analysis of variants using cojo-joint with the same reference panel and excluded variants that were not genome-wide significant in the joint analysis.

### Effect estimates for PRS

A PRS represents an individual's risk of a given disease conferred by the sum of the effects of many common DNA sequence variants. A weight is assigned to each genetic variant based on its strength of association with disease risk. Individuals are then additively scored in a weighted fashion based on the number of risk alleles they carry for each variant in the PRS. To calculate PRS, we used PRScs[14] that calculates posterior SNP effect estimates or weights from original GWAS effect estimates using a Bayesian approach. We used default parameters of PRScs, allowing the method to generate the global shrinkage parameter phi through a Bayesian approach. PRScs use information from an external LD reference panel for this calculation. PRScs calculated posterior SNP effect sizes using a precomputed LD reference panel from EUR ancestry (ldblk_1kg_eur.tar.gz). These effect estimates were used to calculate AAA PRS of individuals in validation cohorts using PLINK2 --score command[52]. Validation cohorts for PRS are described in the Supplementary Note.

### Cox proportional hazards model

We performed another meta-analysis without UKBB by METAL[50], followed by the calculation of PRS weights by PRScs[14], as described above. The UKBB hospital registry data and cause of death data up to March 2020 were used to test the predictive performance of the AAAgen PRS using Cox proportional hazards models. The PRS was adjusted with four first PCs and inverse normalized for the analysis. The analysis only included the white British subset due to the low number of AAA cases in the other ancestries. All variables used in the different models are listed in the Supplementary Table 4b. The baseline of the model was set to the individuals' clinical assessment date. The first occurrence of International Classification of Diseases (ICD)9 codes 441.3 and 441.4 or ICD10 codes I71.3 and I71.4 were recorded together with the date of diagnosis for all AAA cases. Ten-year survival was modeled with follow-up time as the time scale. Prevalent cases ($n = 213$) were excluded from the analyses and individuals who deceased during the follow-up for other causes than AAA were censored. The final analysis included 838 incident AAA cases and 329,983 noncases with median follow-up time of 5.04 and 10.0, respectively. All Cox models were fitted using the R function coxph(). We further tested the models for optimism, and optimism-corrected C-indexes were calculated using R package rms function validate() with method='boot' and 100 repetitions. The model fit of the nested Cox models was compared using likelihood ratio test implemented in R with function anova().

### Gene-set enrichment

We performed gene-set enrichment analysis using DEPICT[15]. DEPICT uses reconstituted gene sets, consisting of 14,462 gene sets obtained from multiple sources and reconstituted using 77,840 publicly available microarray expression datasets. In the reconstituted gene sets, every gene in the human genome is assigned a $z$ score for membership in the set. Using genes from GWAS loci, DEPICT calculates an enrichment $P$ value for reconstituted gene sets. We used AAA summary statistics and clumping with a threshold of $5 \times 10^{-8}$ as input to DEPICT. From the output, gene sets with FDR < 0.01 and $P$ value < $10^{-5}$ are listed in Supplementary Table 5. We identified 17 representative gene sets from this list. Similarity among gene sets (100 genes per gene set) was calculated using the Jaccard index. The similarity matrix was then used to identify the 'exemplar' gene sets by affinity propagation algorithm, implemented in R[53]. These 'exemplar' gene sets are highlighted in Fig. 2a as the representative gene sets.

### Tissue enrichment

Calculation of enrichment in specific tissue types was done by stratified LDSC[16], a method for partitioning heritability. Using GWAS summary

statistics and tissue-specific chromatin marks, this method calculates the enrichment of per-SNP heritability. First, hapmap3 variants with MAF ≥ 0.01 were obtained from the AAA summary statistics using munge_sumstats.py. Then, the analysis was performed using the–-h2-cts option in LDSC. We reported *P* values of enrichment of per-SNP heritability in various chromatin marks from ENCODE and Roadmap Epigenome.

## Cell type enrichment

Processing of scRNA sequencing data from the aorta is described in ref. 17. Calculation of enrichment in specific cell types using scRNA-seq data was performed using RolyPoly[18]. RolyPoly uses a regression-based polygenic model to prioritize relevant cell types from GWAS summary statistics and scRNA-seq data. Average expression per cell type was obtained by using Seurat[54] AverageExpression() function. This generates a matrix with genes as rows and cell types as columns. Columns were normalized using normalize.quantiles(), and rows were normalized using scale() function in R. Because RolyPoly does not work with negative values, we used the absolute values for each entry. To link gene expression with GWAS variants, we used block annotation of 25 kb around the gene's start. For LD statistics, we used 1000 Genomes Phase 3 data precomputed by RolyPoly. We ran the function rolypoly_roll() with 200 bootstrap iterations. *P* values from the bootstrap analyses were reported.

## Identification of protein-altering variants

First, we identified variants that are in LD (>0.8) with index variants using 42,119 unrelated EUR individuals from CHIP + MGI. Variant effect predictor (VEP)[55] was used to predict the molecular consequences of these variants. Genes with at least one variant (index or in LD > 0.8) altering amino acid (missense variant) from VEP output were reported. We observed 21 loci with such genes (Supplementary Table 2). We did not observe any gene with stop-gained or frameshift variants.

## Genes causing related monogenic phenotypes

A list of 53 genes curated in ref. 28 that potentially causes monogenic form of heritable TAA and dissection was incorporated. For a range of related/risk factor phenotypes and disorders (Supplementary Tables 9 and 10), we obtained ClinVar database[56] entries with pathogenic or likely pathogenic variants on 23 June 2021. From these entries, genes (either start or end) that are within 1 Mb of index variant were incorporated. We found 48 loci with at least one gene from these two datasets.

## Polygenic priority score (PoPS) for gene prioritization

We used PoPS[57], a similarity-based gene prioritization method that uses a large set of publicly available RNA sequencing, curated pathway annotation and predicted protein–protein interaction datasets. PoPS was developed with the assumption that causal genes share similar functional characteristics. First, PoPS calculates gene level association statistics from GWAS variant level association statistics and MAGMA gene annotations. Second, features are selected based on precomputed statistics from public resources. Finally, PoPS computes a score for each gene. Based on these scores, for each genome-wide significant locus, we ranked the genes within 1 Mb (either direction) of the index variant and reported the gene with the highest score as the gene prioritized by PoPS. Of 121 loci, 111 loci had at least one gene with a score in the top 10% of PoPS (Supplementary Table 8).

## Expression quantitative trait loci (eQTL) colocalization

We performed colocalization analyses using the R package coloc[58]. Coloc performs an approximate Bayes factor analysis with association statistics. The function coloc.abf() was used to calculate the posterior probabilities for the following: (H0) no association with either trait; (H1/H2) association with one of the two tested traits; (H3) association for both traits but different causal variants and (H4) association for

both traits with the same causal variant. A high posterior probability for H4 (PP4) indicates colocalization of the two trait associations. Colocalization was performed with eQTLs in aorta, liver, whole blood, adipose subcutaneous and visceral omentum from The Genotype-Tissue Expression (GTEx) project v8 (ref. 20). Variants within 500 kb in either direction of GWAS index variants were extracted for the analyses. A threshold of PP4 > 0.5 was used as evidence of colocalization for identifying candidate genes at stage 1 of gene prioritization. This analysis identified 82 loci with at least one gene (Supplementary Table 7). In Supplementary Table 7, we also listed genes with stronger colocalization evidence (PP4 > 0.8).

## TWAS

We used the paradigm of TWAS which performs gene-based association tests. These methods are used to test the association between gene expression predicted by *cis*-eQTLs and phenotype. The MetaXcan package[59] was used to run TWAS with AAA summary statistics. Briefly, we used GWAS tools from the MetaXcan package for summary statistics harmonization and imputation. The imputation step imputes missing GWAS variants using present GWAS variants and the GTEx genotypes. Next, we ran SPrediXcan with the imputed variants and the MASHR expression model (eQTL) of the aorta, liver, whole blood, adipose subcutaneous and visceral omentum from GTEx v8 (ref. 20). For each tissue, significance threshold was decided with correction for multiple testing (0.05/number of genes). This analysis identified 75 loci with at least one gene by TWAS (Supplementary Tables 6 and 7).

## Details of gene prioritization indicators at stage 2

In total, we obtained 523 candidate genes from 121 loci using five indicators at stage 1. At stage 2, these 523 genes were queried in three other datasets to obtain evidence of additional indicators. (1) We used a trio of datasets[60,61] (unpublished data: GSE197748) examining gene transcriptional profiling in two mouse models of AAA (AngII infusion in *ApoE*[−/−] mice and aortic PPE infusion in C57BL/6J mice), specifically those comparing sham/control aortic gene expression versus expression in AAA aortic tissue in experiments featuring 10-week-old male and female mice. Of the 523 candidate genes, 214 demonstrated differential expression in the mouse AAA gene set (Supplementary Table 11). (2) Expression in human abdominal aortic tissue (AAA versus control) from ref. 62 where 79 genes with *q* < 0.05 were included as supported by this indicator (Supplementary Table 12). (3) Related phenotypes observed in mouse knock-outs for candidate genes. We queried data from the Mouse Genome Informatics database and International Mouse Phenotyping Consortium to identify genes for which knockout mice had been reported to have aortic phenotypes consistent with AAA. This generated additional support for five genes in this approach (Supplementary Table 13).

## Gene expression by qPCR

The study was approved by the Ethical Committee of Investigación Clínica del Hospital Santa Creu i Sant Pau. Written informed consent was obtained from all patients. The study conformed to the principles outlined in the Declaration of Helsinki. All patients underwent surgery at Hospital de la Santa Creu i Sant Pau. Samples were obtained from the remaining mid-infrarenal aortic wall after exclusion and prosthetic replacement of AAA. Normal aortas were obtained from healthy aorta from multiorgan donors, and samples were also taken from the midportion of the infrarenal abdominal aorta during organ collection. When a luminal thrombus was present, it was separated before the aorta biopsy was taken and aortic tissue was washed twice with cold PBS. A portion of each sample was placed in an RNAlater solution (Qiagen GmbH) and stored at 4 °C for 24 h before long-term storage at −80 °C until further processing for RNA isolation. Further information can be found in ref. 21. Tissues were homogenized in the FastPrep-24 homogenizer and Lysing Matrix D tubes (MP Biomedicals). RNA was extracted using Trizol

(Invitrogen) following the manufacturer's instructions. cDNA was prepared by reverse transcribing 1 μg RNA with a High-Capacity cDNA Archive Kit with random hexamers (Applied Biosystems). mRNA expression of the selected genes was studied by real-time PCR in an ABI Prism 7900HT using predesigned validated assays (TaqMan Gene Expression Assays; Applied Biosystems) and universal thermal cycling parameters. Relative expression was expressed as transcript/β-actin ratios.

### Genetic correlation
Genetic correlations between AAA and other traits were calculated using cross-trait LDSC[63]. This method relies on GWAS summary statistics to estimate the genetic correlation between pairs of traits. The GWAS summary statistics were filtered to identify genetic variants included in the HapMap3 reference panel. Cross-trait LDSC was then performed using ldsc v.1.0.1 using the default EUR ancestry LD panel.

### PheWAS analysis
We performed a logistic-regression-based PheWAS of 121 independent variants against the entirety of the MRC-IEU open GWAS project[29]. The full summary statistics were then filtered to exclude the following phenotypes: methylation quantitative trait locus (mQTLs), pQTLs, eQTLs, sex-specific GWAS and non-EUR and nonmixed ancestry GWAS. Furthermore, the full summary statistics were filtered to GWAS studies that contained 50 or more cases. When the trait was measured at multiple locations (that is, right and left arm), only results from one side were queried. For replicate phenotypes, the largest study of that phenotype was chosen. PheWAS results for each independent variant were considered significant if they passed a Bonferroni-corrected threshold for 161,535 (121 variants and 1,335 phenotypes) tests conducted. After calculating the node and edge maps using NETMAGE[64], only connections that represent a weight of >3 SNPs were used to make the graph more interpretable. Raw network statistics were calculated in Gephi[65] including modularity or cluster ID using details in ref. [66]. This method works to create modules that minimize the number of edges that enter and leave the module. The granularity of the definition of the modules or resolution can be adjusted, with lower resolution generating a higher number of modules. Our optimization efforts led us to set the resolution parameter as 0.561 generating seven distinct modules. Colocalization analysis with the largest available GWAS of lipid[31], blood pressure[32] and smoking[33] traits was performed by coloc[58], as described in the eQTL colocalization section.

### Mendelian Randomization (MR)
Genetic associations between lipoprotein fractions (exposure) and AAA outcome were tested initially using inverse-variance weighted MR for a single lipid exposure and then using the MR-BMA methodology for multivariable models[34]. MR-BMA is an extension of multivariable MR using a Bayesian variable selection method in an effort to identify likely causal risk factors among correlated exposures. In the primary analysis, the instrumental variables consisted of independent genetic variants ($r^2 < 0.001$ based on 1000 Genomes[67] EUR ancestry Reference Panel) associated with any major lipoprotein-related traits (ApoB, LDL-C, HDL-C, TG, ApoA1 and non-HDL-C) at genome-wide significance in the UKBB based on 361,194 EUR-ancestry participants as previously described[68] (Supplementary Table 23). Given the substantial correlation between the non-HDL-C and LDL-C lipid fractions ($r^2 > 0.9$), the following two separate models were used for analysis: one containing LDL-C, HDL-C, ApoA1, ApoB and triglycerides ($n = 519$ independent genetic variants), and another substituting non-HDL-C for LDL-C ($n = 450$ independent genetic variants; Fig. 6a). The subsequent MR-BMA analysis was completed using AAA GWAS summary statistics from the current study. Variable selection was based on marginal inclusion probabilities for which an empirical permutation procedure was used to derive $P$ values. The Nyholt procedure of effective tests was used to account for the strong correlation among the lipoproteins, with

a multiple testing–adjusted $P$ value of $P = 0.05$ set as the significance threshold[69]. Further details of statistical analysis are described in the Supplementary Note.

### Proteome-wide MR
To identify possible therapeutic targets for AAA, we performed a proteome-wide MR analysis using high-confidence *cis*-acting genomic instruments for circulating plasma proteins passing pleiotropy and consistency filters as previously described[70]. In brief, variants were selected that were associated with any protein at genome-wide significance ($P < 5 \times 10^{-8}$). For this analysis, we focused only on *cis*-pQTLs previously classified as having the highest relative level of reliability ('Tier 1' as defined in ref. [70]). After excluding the major histocompatibility complex region, LD clumping to identify instruments composed of independent pQTLs was performed using the TwoSampleMR package[71] with an $r^2$ threshold of <0.001. After restricting to 717 circulating proteins with overlapping proteomic and genomic data (Supplementary Table 27), we performed a pQTL screen as follows: (1) for exposure–outcome pairs with two or more available genetic instruments as proxies for the exposure, inverse-variance weighted MR was performed; (2) when only a single genetic proxy for the exposure was present, Wald-ratio MR was performed. We set a Benjamini–Hochberg FDR < 0.05 for statistical significance. For each protein–AAA association in *cis*-pQTL-MR above, we performed colocalization to provide supporting evidence for causal associations among traits. Colocalization was performed using HyPrColoc[72].

### *Pcsk9*$^{-/-}$ mice and AAA progression
All animal protocols were approved by the Administrative Panel on Laboratory Animal Care at Stanford University (http://labanimals.stanford.edu/) and the VA Palo Alto Health Care System Institutional Animal Care and Use Committee and followed the National Institutes of Health and US Department of Agriculture Guidelines for Care and Use of Animals in Research. Male mice (wild-type and *Pcsk9*$^{-/-}$ mice on a C57BL/6J background) were purchased from the Jackson Laboratory.

### PPE infusion model
The PPE infusion model to induce mouse AAA was performed as previously described at 10 weeks of age[39]. The proximal and distal aorta were temporarily ligated or clamped, followed by an aortotomy above the iliac bifurcation. A catheter was used to infuse the aorta for 5 min at 120 mmHg with saline containing type I PPE (2.5 U ml$^{-1}$; Sigma-Aldrich), and the aortotomy was then repaired. The induced AAA aortic segment (between the left renal artery and the bifurcation) was collected at 28-d postsurgery.

### Lipid measurements
Plasma cholesterol, LDL and HDL levels were measured at Stanford Animal Diagnostics Laboratory on the Siemens Dimension EXL200/LOCI analyzer.

### Aortic diameter measurements by ultrasound imaging
At baseline and 3, 7, 14, 21 and 28 d after aneurysm induction, B-mode ultrasound imaging was performed on the operated mice to assess the AAD as previously described[39].

### Mouse AAA model statistical analysis
Data are presented as means ± s.e.m. Level of significance was determined using the Mann–Whitney test. Shapiro–Wilk test was performed to test normality. A $P$ value < 0.05 was considered statistically significant.

### Reporting summary
Further information on research design is available in the Nature Portfolio Reporting Summary linked to this article.

## Data availability

Meta-analysis summary statistics and PRS weights are available at https://csg.sph.umich.edu/willer/public/AAAgen2023/. Unpublished mouse transcriptome data are available at Gene Expression Omnibus (GSE197748). Due to stipulations of the IRB, AAA bulk RNA-seq data are only available directly from the authors (isurakka@med.umich. edu) and will be provided to qualified investigators with appropriate IRB approval and materials transfer agreement.

## Code availability

Software tools used for discovery cohort GWAS are outlined in the Supplementary Note. Publicly available software tools were used to perform meta-analysis and downstream analysis. These tools include METAL 2011-03-25 (http://csg.sph.umich.edu/abecasis/Metal/), PRScs v1.0.0−April 11, 2020 (https://github.com/getian107/PRScs), GCTA 1.92.1 (https://yanglab.westlake.edu.cn/software/gcta/), DEPICT version 1 rel194 (https://github.com/perslab/depict), LDSC v1.0.0 (https://github.com/bulik/ldsc), RolyPoly (https://github.com/dcalderon/rolypoly), VEP (https://useast.ensembl.org/info/docs/tools/vep/), PoPS v0.2 (https://github.com/FinucaneLab/pops), Coloc v3.2.1 (https://cran.r-project.org/web/packages/coloc/), MetaXcan v0.7.5 (https://github.com/hakyimlab/MetaXcan), Gephi (https://github.com/gephi/gephi), MR-BMA v2021-10-05 (https://github.com/verena-zuber/demo_AMD), TwoSampleMR v0.5.6 (https://mrcieu.github.io/TwoSampleMR/) and HyPrColoc v2021-07-23 (https://github.com/jrs95/hyprcoloc).

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

## Acknowledgements

This work was supported by funding from Health Research Council of New Zealand (awards 14/155, 17/402, 20/144 to G.T.J.), British Heart Foundation (grants CS/14/2/30841 and RG/18/10/33842 to M.B.), Veterans Administration Office of Research and Development (I01-BX003362) and Tobacco-Related Disease Research Program (T29IR0636 to P.S.T.), the National Institutes of Health (NIH; R35-HL135824-03 and R01-HL142023-02 to C.J.W., R01-HL166991 to S.M.D.). A.H.C. is supported by the Independent Research Fund Denmark (0134-00363B) and the Novo Nordisk Foundation (NNF20OC0065799). C.A.H. is supported by the NIH/National Heart, Lung and Blood Institute (NHLBI) Institutional Training Grant (T32HL098049) and the Propel Postdoctoral Scholars Program of the Stanford University School of Medicine. C.G. is funded by the British Heart Foundation under grants CS/14/2/30841 and RG/18/10/33842. D.G. is supported by the British Heart Foundation Center of Research Excellence (RE/18/4/34215) at Imperial College. D.K. is supported by the Department of Veterans Affairs (IK2BX005759-01), the American Heart Association (23SCEFIA1153369), and the Baszucki Research Initiative provided to Stanford Vascular Surgery. J.M.S. is supported by the Tobacco-Related Disease Research Program (T31IR1845). I.J.K. is funded by grants HG06379 and HG11710 from the National Human Genome Research Institute and K24HL137010 from the NHLBI. K.B. and S.B. acknowledge the Novo Nordisk Foundation (grants NNF17OC0027594 and NNF14CC0001). K.L.S. is supported by Wellcome Trust Doctoral Training Program reference 222959/Z/21/Z. M.S.-L. is supported by a Miguel Servet contract from the ISCIII Spanish Health Institute (CP17/00142) and cofinanced by the European Social Fund. M.G.L. is supported by the Institute for Translational Medicine and Therapeutics of the Perelman School of Medicine at the University of Pennsylvania and the NIH/NHLBI National Research Service Award postdoctoral fellowship (T32HL007843). O.D. is supported by the Mayo Clinic Clinician Investigator training program. S.W.v.d.L. is funded through EU H2020 TO_AITION (grant 848146), has received Roche funding for unrelated work and would like to thank the support of the

Netherlands CardioVascular Research Initiative of the Netherlands Heart Foundation (CVON 2011/B019 and CVON 2017-20: generating the best evidence-based pharmaceutical targets for atherosclerosis (GENIUS I&II)), the ERA-CVD program 'druggable-MI-targets' (grant 01KL1802) and the Leducq Foundation 'PlaqOmics'. S.B. is supported by a Sir Henry Dale Fellowship jointly funded by the Wellcome Trust and the Royal Society (204623/Z/16/Z). This research was funded by the United Kingdom Research and Innovation Medical Research Council (MC_UU_00002/7) and supported by the National Institute for Health Research Cambridge Biomedical Research Center (BRC-1215-20014). The views expressed are those of the authors and not necessarily those of the National Institute for Health Research or the Department of Health and Social Care. S.K.G. is supported by NIH (grant R35HL161016), the Department of Defense and the University of Michigan A. Alfred Taubman Institute. S.M.D. is supported by a Career Development Award (IK2-CX001780) from the US Department of Veterans Affairs Clinical Science Research and Development Service. Y.H.R. is supported by the Dean's Postdoctoral Fellowship from the Stanford University School of Medicine. Y.E.C. is supported by NIH (grant R01-HL109946). This publication does not represent the views of the Department of Veteran Affairs or the US Government.

## Author contributions

T.R., D.K., M.G.L., I.S., G.T.J., M.J.B., P.S.T., C.J.W. and S.M.D. were responsible for the study design, interpretation of results and drafting of the manuscript. I.S., G.T.J., M.J.B., P.S.T., C.J.W. and S.M.D. oversaw the study's supervision and funding. T.R., D.K., M.G.L., J.M.S., N.L.T., C.G., V.Z., F.S., S.B., D.G., M.S.-L., L.C.T., K.G., I.S., G.T.J., M.J.B., P.S.T., C.J.W. and S.M.D. conducted the meta-analysis, QC and downstream analysis. J.M.S., Y.H.R., A.D., C.A.H. and P.S.T. performed the mouse experiment. T.R., D.K., C.G., W.E.H., I.H.L., S.S.V., A.E.L., G.E., G.T., S.E.G., O.D., J.W.P., R.L.J., F.P.-V., J.B.N., B.N.W., B.M.B., J.D., O.P., L.C.J., T.L.E., D.L., K.B., S.B., R.L.J., M.T.G.-B., J.Z., L.M.R., R.L., A.H., A.W., K.M., J.S.L., L.M.O., E.S., G.O., C.P.N., K.L.S., J.A.v.H., S.W.v.d.L., J.v.S., M.C., F.M.D., R.W., J.E.G., J.L.E., D.M.C., P.K.H., S.K.G., Y.E.C., W.G., J.S.P., N.P., O.B.P., CE., W.T., K.H., D.G., S.G., U.T., H.H, K.S., M.A.F., A.B., I.J.K., M.D.R., A.H.C., K.K.I., N.E., H.S., S.R.O., H.B., H.U., M.S.-L., I.S., G.T.J., M.J.B., P.S.T., C.J.W. and S.M.D. were responsible for the data collection and analysis of the discovery cohort. All authors were involved in editing and the final review of the manuscript.

## Competing interests

A.E.L. is an employee of Regeneron Genetics Center and a shareholder of Regeneron Pharmaceuticals. A.B. is an employee of Regeneron Genetics Center and a shareholder of Regeneron Pharmaceuticals. C.J.W. is currently employed by Regeneron Pharmaceuticals, but scientific input to this manuscript occurred prior to employment at Regeneron Pharmaceuticals. D.K. is a scientific advisor and reports consulting fees from Bitterroot Bio, Inc unrelated to the present work. D.L. is an employee of Regeneron Genetics Center and a shareholder of Regeneron Pharmaceuticals. D.G. is employed part-time by Novo Nordisk. J.B.N. is employed by Regeneron Pharmaceuticals, unrelated to this work. J.A.v.H. is a consultant and/or proctor for Terumo Aortic, Cook Medical, Microport, WL Gore and Philips. M.A.F. is an employee of Regeneron Genetics Center and a shareholder of Regeneron Pharmaceuticals. M.R. is a member of the Scientific Advisory Board for Cipherome. S.E.G. is currently employed by Regeneron Pharmaceuticals. S.M.D. receives research support to the University of Pennsylvania from RenalytixAI and Novo Nordisk and personal consulting fees from Calico Labs, all outside the scope of the current research. S.B. has ownerships in Intomics A/S, Hoba Therapeutics Aps, Novo Nordisk A/S, Lundbeck A/S, ALK-Abello A/S and managing board memberships in Proscion A/S and Intomics A/S. The remaining authors declare no competing interests.

## Additional information

**Correspondence and requests for materials** should be addressed to Tanmoy Roychowdhury, Cristen J. Willer or Scott M. Damrauer.

Cristen J Willer
Tanmoy Roychowdhury

# Reporting Summary

## Statistics

For all statistical analyses, confirm that the following items are present in the figure legend, table legend, main text, or Methods section.

| n/a | Confirmed | |
|---|---|---|
| ☐ | ☒ | The exact sample size (*n*) for each experimental group/condition, given as a discrete number and unit of measurement |
| ☐ | ☒ | A statement on whether measurements were taken from distinct samples or whether the same sample was measured repeatedly |
| ☐ | ☒ | The statistical test(s) used AND whether they are one- or two-sided *Only common tests should be described solely by name; describe more complex techniques in the Methods section.* |
| ☐ | ☒ | A description of all covariates tested |
| ☐ | ☒ | A description of any assumptions or corrections, such as tests of normality and adjustment for multiple comparisons |
| ☐ | ☒ | A full description of the statistical parameters including central tendency (e.g. means) or other basic estimates (e.g. regression coefficient) AND variation (e.g. standard deviation) or associated estimates of uncertainty (e.g. confidence intervals) |
| ☐ | ☒ | For null hypothesis testing, the test statistic (e.g. *F*, *t*, *r*) with confidence intervals, effect sizes, degrees of freedom and *P* value noted *Give P values as exact values whenever suitable.* |
| ☐ | ☒ | For Bayesian analysis, information on the choice of priors and Markov chain Monte Carlo settings |
| ☐ | ☒ | For hierarchical and complex designs, identification of the appropriate level for tests and full reporting of outcomes |
| ☐ | ☒ | Estimates of effect sizes (e.g. Cohen's *d*, Pearson's *r*), indicating how they were calculated |

*Our web collection on statistics for biologists contains articles on many of the points above.*

## Software and code

Policy information about availability of computer code

| Data collection | No software was used for data collection. |
|---|---|
| Data analysis | Software tools used for discovery cohort GWAS are outlined in the Supplementary Note. Publicly available software tools were used to perform meta-analysis and downstream analysis. These tools include METAL 2011-03-25 (http://csg.sph.umich.edu/abecasis/Metal/), PRS-cs v1.0.0-Apr 11, 2020 (https://github.com/getian107/PRScs), GCTA 1.92.1 (https://yanglab.westlake.edu.cn/software/gcta/), DEPICT version 1 rel194 (https://github.com/perslab/depict), LDSC v1.0.0 (https://github.com/bulik/ldsc), RolyPoly (https://github.com/dcalderon/rolypoly), VEP (https://useast.ensembl.org/info/docs/tools/vep/), PoPS v0.2 (https://github.com/FinucaneLab/pops), Coloc v3.2.1 (https://cran.r-project.org/web/packages/coloc/), MetaXcan v0.7.5 (https://github.com/hakyimlab/MetaXcan), Gephi (https://github.com/gephi/gephi), MR-BMA v2021-10-05 (https://github.com/verena-zuber/demo_AMD), TwoSampleMR v0.5.6 (https://mrcieu.github.io/TwoSampleMR/), HyPrColoc v2021-07-23 (https://github.com/jrs95/hyprcoloc). |

For manuscripts utilizing custom algorithms or software that are central to the research but not yet described in published literature, software must be made available to editors and reviewers. We strongly encourage code deposition in a community repository (e.g. GitHub). See the Nature Portfolio guidelines for submitting code & software for further information.

## Data

Policy information about <u>availability of data</u>

All manuscripts must include a <u>data availability statement</u>. This statement should provide the following information, where applicable:
- Accession codes, unique identifiers, or web links for publicly available datasets
- A description of any restrictions on data availability
- For clinical datasets or third party data, please ensure that the statement adheres to our <u>policy</u>

Meta-analysis summary statistics and PRS weights are available here: https://csg.sph.umich.edu/willer/public/AAAgen2023/. Unpublished mouse transcriptome data is available at GEO (GSE197748). Due to stipulations of the IRB, abdominal aortic aneurysm bulk RNA-seq data is only available directly from the authors (isurakka@med.umich.edu) and will be provided to qualified investigators with appropriate IRB approval and materials transfer agreement.

## Human research participants

Policy information about <u>studies involving human research participants and Sex and Gender in Research.</u>

| | |
|---|---|
| Reporting on sex and gender | The results are not sex-stratified. As majority of AAA cases occurs in the male sex, dataset primarily consists of male sex. GWAS analysis in discovery cohorts incorporated sex as a covariate. |
| Population characteristics | The meta-analysis incorporates data from individuals of European and African ancestry. Following covariates were used for discovery cohort GWAS: ARIC (Age, Sex, Pack years of smoking, PCs 1-5); CHB-CVDC+DBDS (Birth year, Sex, PCs 1-10); CHIP+MGI (Birth year, Sex, Array version, PCs 1-4 ); deCODE (Age, Age2, Sex, County of origin, Some additional); DiscovEHR (Age, Age2, Sex, AgexSex, Age2xSex, PCs1-10); eMERGE (Age, Sex, Clinical site, PCs1-5); HUNT (Birth year, Sex, Batch, PCs 1-4 ); Mayo VDB (Study enrolment age, Sex, Genotyping platform, PCs 1-5); MVP (Age, Sex, PCs 1-5); NZ (NA); PMBB (Age at recruitment, Sex, Genetic determined ancestry, PCs 1-10); TABS (Birth year, Sex, Batch, PCs 1-4); UKAGS+VIVA (Birth year, Sex, Batch, PCs 1-4); UKBB (Age matching for controls, Sex). These information are also listed in Supplementary Table 1. |
| Recruitment | Outlined in Supplementary Methods relevant to discovery cohorts. |
| Ethics oversight | Outlined in Supplementary Methods relevant to discovery cohorts. |

Note that full information on the approval of the study protocol must also be provided in the manuscript.

# Field-specific reporting

Please select the one below that is the best fit for your research. If you are not sure, read the appropriate sections before making your selection.

☒ Life sciences　　　☐ Behavioural & social sciences　　　☐ Ecological, evolutionary & environmental sciences

For a reference copy of the document with all sections, see <u>nature.com/documents/nr-reporting-summary-flat.pdf</u>

# Life sciences study design

All studies must disclose on these points even when the disclosure is negative.

| | |
|---|---|
| Sample size | Sample size was determined based on using all genetic data (European and African ancestry) available from 14 discovery cohorts. Participants were excluded if they failed to meet case or control definitions. |
| Data exclusions | Data were excluded if they did not pass our QC metrics. |
| Replication | External datasets lack power for replication. We demonstrated consistent effect size estimation (Supplementary Figure 7, Supplementary Table 2) using Finngen and PRS validation cohorts that were not incorporated in the meta-analysis (P<0.05 in 80/121 loci). |
| Randomization | Randomization is not relevant for this binary trait GWAS study which is a retrospective analysis of AAA cases and controls. |
| Blinding | Randomization is not relevant for this binary trait GWAS study which is a retrospective analysis of AAA cases and controls. |

# Reporting for specific materials, systems and methods

We require information from authors about some types of materials, experimental systems and methods used in many studies. Here, indicate whether each material, system or method listed is relevant to your study. If you are not sure if a list item applies to your research, read the appropriate section before selecting a response.

## Materials & experimental systems

| n/a | Involved in the study |
|-----|----------------------|
| ☒ ☐ | Antibodies |
| ☒ ☐ | Eukaryotic cell lines |
| ☒ ☐ | Palaeontology and archaeology |
| ☐ ☒ | Animals and other organisms |
| ☒ ☐ | Clinical data |
| ☒ ☐ | Dual use research of concern |

## Methods

| n/a | Involved in the study |
|-----|----------------------|
| ☒ ☐ | ChIP-seq |
| ☒ ☐ | Flow cytometry |
| ☒ ☐ | MRI-based neuroimaging |

# Animals and other research organisms

Policy information about <u>studies involving animals</u>; <u>ARRIVE guidelines</u> recommended for reporting animal research, and <u>Sex and Gender in Research</u>

| Laboratory animals | 10 week-old male C57Bl/6 and PCSK9 (-/-) mice on a C57Bl/6 background were used in the current study. Animals were housed in the animal facility at the VA Palo Alto Health Care System which was temperature (68-72 F)- and humidity (70%)-controlled under a 12-h light/dark cycle. |
|---|---|
| Wild animals | No wild animals were used in this study. |
| Reporting on sex | Male mice were used in this study as the majority of AAA cases occurs in the male sex. |
| Field-collected samples | No field collected samples were used in this study. |
| Ethics oversight | All animal protocols were approved by the Administrative Panel on Laboratory Animal Care at Stanford University (http://labanimals.stanford.edu/) and the VA Palo Alto Health Care System Institutional Animal Care |

Note that full information on the approval of the study protocol must also be provided in the manuscript.

