## [Peer Review File · Nature Genetics]

Peer Review Information

Manuscript Title: Genome-wide association meta-analysis identifies risk loci for abdominal aortic aneurysm and highlights PCSK9 as a therapeutic target

Corresponding author name(s): Dr Scott (M) Damrauer Dr Cristen (J) Willer Dr Tanmoy Roychowdhury

Editorial Notes:

Transferred manuscripts This document only contains reviewer comments, rebuttal and decision letters for versions considered at Nature Genetics

Reviewer Comments & Decisions:

Decision Letter, initial version:

10th Aug 2022

Dear Dr Damrauer,

Your Article, "Multi-ancestry GWAS deciphers genetic architecture of abdominal aortic aneurysm and highlights PCSK9 as a therapeutic target" has now been seen by 3 referees. You will see from their comments below that while they find your work of interest, some important points are raised. We are interested in the possibility of publishing your study in Nature Genetics, but would like to consider your response to these concerns in the form of a revised manuscript before we make a final decision on publication.

To guide the scope of the revisions, the editors discuss the referee reports in detail within the team with a view to identifying key priorities that should be addressed in revision. In this case, we think all three referees have provided constructive reviews aimed at strengthening the analyses and improving the presentation. We particularly ask that you address the major concerns regarding the gene prioritization section (Reviewer #2) and the methods and interpretation of the results (Reviewer #3). In addition, you may need to carefully check the use of the term "multi-ancestry" and tone down some claims wherever appropriate. Please address all their comments as thoroughly as possible with appropriate revisions. We hope that you will find the prioritized set of referee points to be useful when revising your study.

We therefore invite you to revise your manuscript taking into account all reviewer and editor comments. Please highlight all changes in the manuscript text file. At this stage we will need you to upload a copy of the manuscript in MS Word .docx or similar editable format.

*2) If you have not done so already please begin to revise your manuscript so that it conforms to our Article format instructions, available

[here](http://www.nature.com/ng/authors/article_types/index.html).

*3) Include a revised version of any required Reporting Summary:

[redacted]

We hope to receive your revised manuscript within 3 to 6 months. If you cannot send it within this time, please let us know.

Nature Genetics is committed to improving transparency in authorship. As part of our efforts in this direction, we are now requesting that all authors identified as 'corresponding author' on published papers create and link their Open Researcher and Contributor Identifier (ORCID) with their account on the Manuscript Tracking System (MTS), prior to acceptance. ORCID helps the scientific community achieve unambiguous attribution of all scholarly contributions. You can create and link your ORCID from the home page of the MTS by clicking on 'Modify my Springer Nature account'. For more

information please visit please visit www.springernature.com/orcid.

Sincerely,
Wei

Wei Li, PhD
Senior Editor
Nature Genetics
New York, NY 10004, USA
www.nature.com/ng

Reviewers' Comments:

Reviewer #1:

Remarks to the Author:

Roychowdhury et al. performed a meta-analysis of GWAS for abdominal aortic aneurysms using the largest sample size to date (39,221 cases), including European and African populations, and identified 97 novel loci and 144 independent signals. The PRS built on the summary statistics of this GWAS performed better than the one derived from the GWAS of Klarin et al. reported in 2020 (7,642 cases). The GWAS summary statistics and gene expression data also pinpointed genes, molecular pathways, and cell types that were prioritized for the disease. The genetic relationship with thoracic aortic aneurysms was also presented. Finally, using PheWAS, MR, and a mouse model, the authors proposed that PCSK9 inhibitors may be effective in the treatment of AAA. The downstream analysis is up-to-date and it is interesting to present the possibility of a new therapeutic agent. There are several suggestions and points of clarification I would like to make:

1. In the studies used in the meta-analysis of the AAA discovery cohort (Figure S1 and Table S1), there were some studies with high or low lambda GC. Could you justify these inflated or deflated values? Especially, in order to interpret these values accurately (in particular, whether studies with high lambda GC were affected by polygenicity or not), LDSC intercept values may be helpful.

2. How much of the heritability can be explained by this AAA GWAS?

3. while I understood the usefulness of internal replication for novel loci, is external replication not possible?

Finngen published summary statistics for AAA (https://r7.finngen.fi/pheno/I9_ABAORTANEUR)

4. Main text, lines 203-212. Regarding the performance of PRS, just the values of Harrel's C-index were used to discuss whether the performance was better or not. However, you should test whether the difference was statistically significant using a method such as Bootstrap or something like that.

5. Main text, line 214-. the DEPICT algorithm prioritized pathways, whereas the pathways available in the default setting were hard to interpret because there were so many similar ones. After clustering the pathways that exhibit similar functionality and obtaining the distinguished clusters (which include an 'exemplar' pathway and member pathways), could you perform the enrichment analysis using only the 'exemplar' pathways?

6. Main text, lines 219-234. The authors performed cell-type enrichment analysis using scRNA data from the aorta. On the other hand, in usual tissue enrichment analysis, which organs/ tissues are enriched with GWAS signals? What cell types are enriched in organs/ tissues associated with AAA other than the aorta?

7. Main text, line 241. 4 tissues: aorta, liver, whole blood, and adipose.
While I understand that the aorta was included, what criteria were used to select the other three?

8. Gene prioritization. Gene-prioritization strategies (main text, line 235-) identified putative causal genes at the AAA risk loci, whereas the integration with external datasets (main text, line 298-) elucidated AAA-associated genes in the aorta. This is confusing because both utilized TWAS, mouse AAA model, and human AAA gene expression data. Can you please sort this out?

9. Main text, lines 324-325. Showing the genetic correlations using GWAS summary statistics of AAA, TAA, and lipid traits could be useful to understand the context. I think you can use the European LD structure to calculate it since Europeans are the majority despite a multi-ancestry GWAS.

10. Main text, line 368-370. The authors have checked major clinical risk factors for AAA using pairwise colocalization. Again, could you show us the genetic correlations R_g between these diseases? Additionally, diabetes mellitus is known to be negatively associated with AAA or TAA (PMID: 30052821). How about the association between your AAA result and diabetes? Also, how about the association with other atherosclerotic cardiovascular diseases? (e.g. coronary artery disease)

11. Main text, lines 392-395. In MR-BMA analysis, instrumental variables were extracted from UK biobank-based lipid GWASs. Because your GWAS included UKBB samples, the assumption of two-sample MR that independent GWASs should be used for exposure and outcome was violated. Could you explain more about that?

12. Main text, line 437-. Pcsk9^{-/-} mouse model. The authors mentioned that the PCSK9^{-/-} mouse model exhibited suppressed progression of AAA, in which how were the gene expression alterations of AAA in the mice? I think further molecular biological mechanisms can be identified by analyzing this point.

Also, PCSK9 inhibition reduces lipid levels. To what extent were serum lipids reduced in the pcsk9^{-/-} mice? If the effect of PCSK9 inhibition was lipid-mediated, would lowering lipids by methods other than PCSK9 inhibition also strongly suppress AAA?

13. PCSK9 inhibitors are one of the options for lipid-lowering therapy in the presence of atherosclerotic diseases. It would be interesting to explain a bit more about the relationship between AAA and other atherosclerotic diseases via PCSK9.

Minor comments

1. GO analysis (Table S15). GO proposes molecular functions, biological processes, and cellular components. Where is the biological process?
2. Manhattan Plot (Figure S3). It will be easier to understand if you can plot the new loci in a different color. Please also provide the name of the novel loci.
3. Figure S4, S5. If the range of 95% confidence interval is shown, we can understand how accurate the effect estimates were.
4. "FDR P-value" is an odd expression and should simply be "FDR" or "q-value".
5. Main text, line 1073. The expression "Welch's t-test for parametric and Mann-Whitney for non-parametric". This is an odd expression. The t-test is a parametric test and assumes a normal distribution, while the Mann-Whitney test is a nonparametric test and is used when a normal distribution cannot be assumed.

Reviewer #2:
Remarks to the Author:
Roychowdhury et al

This paper describes the results of the largest GWAS to date for AAA – with 97 new loci. The authors have performed comprehensive downstream analyses with the new findings providing new insights into underlying disease biology, and demonstrate overlap with other aortopathies. The PRS work can explain more than the clinical risk factors and they highlight results to indicate that PCSK9 is a therapeutic target.

The authors have done a tremendous amount of work, and the results contribute new knowledge to the causes of AAA with discovery of new loci for this condition and highlight some therapeutic options for consideration.

I do have some specific questions which I will list below.

1. The description of the meta-analysis was clear to follow and nice to see the internal validation you did. You indicate removal of rare variants, from looking at the individual studies it does not look like a MAF threshold was used? Therefore, it was a little odd to take variants all the way forward and then remove, as the variant passed internal validation – or did it, this is not clear from text or suppl. Table 2.
2. You performed several PRS analysis – the motivation for all the different analyses was not really described – you would have predicted your new PRS would outperform the Klarin one, I thought the text in this section could be reduced.
3. The gene prioritisation section I found difficult to follow and long. I appreciate all the analyses you did but it is not clear at the end which genes have strong evidence and from which sources. I appreciate creating a way to prioritise candidate genes is quite subjective and across GWAS studies there is no wrong or right approach. What you have done however does need to be clearer. I wondered if you can provide an overview figure of what was done here for the reader? At the outset

- you do describe 8 different indicators, so at the outset what was the starting number of candidate genes? You move to then describing 523 genes with 5 of the above indicators, so there are 523 genes which have support from 5/8 indicators. I got more confused when you started with describing results following rules again in the second last paragraph starting with "next we used the following rules". There is no page numbers for me to reference. I think the whole section on gene prioritisation needs a rethink in reporting so it is clear what the input was for testing for enrichment with GO. Once this section is more clearly described, then highlight as you have done interesting pathways and genes.
4. I wondered for formally assessing the overlap with thoracic aortic aneurysm if you should consider using pairwise- GWAS (<https://github.com/joepickrell/gwas-pw>)?
 5. I thought once the gene prioritisation section was clearer, the section labelled "integration with external datasets elucidates..." Could follow this, the overlap with other traits follows this section not precedes as it currently stands.
 6. In the section "pleiotropic AAA risk..." I did wonder if you also wished to state a little more on the relationship with BP as well?
 7. The MR section and leveraging potential drug purposing opportunities I thought was interesting and the results from the mouse model. I wondered if this work can be highlighted a little more in the abstract. You provide lots of text on genes and pathways, these are results of equivalent interest to readers with the availability of PCSK9 inhibitors.
 8. The discussion was strong and great on the limitations.

Minor points

1. Add a Table of Contents for the Suppl. Tables.
2. Figure 3 is great and summarises the results nicely, I would expand in the main text describing this figure when you reconfigure reporting of your gene prioritisation pipeline.
3. Figure 5 – figures and text needs to be bigger on axes
4. Figure 7 – can you expand the legend to indicate source of your categories for the genes from the suppl. Tables.
5. Suppl Table 14 is great, can you provide some text as a legend to explain the final gene selection in column AC

Reviewer #3:

Remarks to the Author:

Roychowdhury et al. have performed a GWAS meta-analysis in AAA and followed up the findings with downstream functional and expression analysis. While the results are impressive there are some major concerns regarding the methods and interpretation of the results.

1. I found the use of the term "multi-ancestry" in the title and abstract quite misleading. Apart from the very small fraction of African Americans included in the meta-analysis, there is no other ancestries included, no ancestry-specific analysis, no estimation of the gains in PRS performance or fine-mapping improvement from the addition of non-European ancestries. I do not consider this study as multi-ancestry in that respect. All downstream analysis are also limited to European reference panels or annotation features.
2. It is not very clear whether this meta-analysis was based on a uniform analysis plan for all contributing studies. There is a range of different covariates used from the studies that do not necessary fall under the cohort-specific population stratification adjustments. For example ARIC is adjusting for smoking, DiscovEHR for a series of age and gender combinations, eMERGE only for PCS,

there is no mention of covariates for NZ AAA Genetics studies etc. It would be helpful to supply a supplementary table with cohort-specific analytical details. Do the authors fear that this lack of standardisation in the analysis might have impacted the results?

3. The use of UK Biobank in this meta-analysis and subsequent analysis is very difficult to follow. I suggest the authors provide with a diagram for that. My main questions are: 1. Why are there only 6,276 controls in the discovery stage and how were these selected? 2. When testing the PRS in the UKAGS-UKB validation study it is not clear if the PRS is with or without UK Biobank, 3. Was the MR performed in the UK Biobank with the meta-analysis summary statistics with or without UK Biobank, 4. Please clarify the adjustment used for the association analysis and whether relatedness was considered, 5. Please clearly mention in the main text the number of incident AAA cases in UK Biobank.

4. The MVP study seems to have disproportionately higher number of AAA cases compared to Biobanks of similar size (eMERGE or UK Biobank). How do the authors explain that? Could that be due to misclassification?

5. The authors perform the meta-analysis and then use the same meta-analysis without the MVP as a replication step. This might be considered a sensitivity analysis but not replication. The authors should use the PRS validation studies for replicating their GWS findings.

6. The authors claim 121 independent loci but these are only identified by distance pruning, therefore not tested for independence. The authors should apply the cojo-joint in GCTA to assess independence. On a similar note, it is not clear why the authors use the cojo-cond for identifying the additional 24 secondary loci, as the cojo-joint is a better approximation of the exact conditional analysis. Please also provide joint p-values in the supplementary table.

7. Please add the covariates used in the cox model methods section.

8. The PRS computation is not clearly described. Please mention which tuning criteria were used and how you chose the best performing PRS. Did you test it in an independent study?

9. Figure S6. The ORs for the current PRS are unrealistic and so are the CIs, especially for the right-hand side deciles. It seems unlikely that the validation studies have the statistical power to detect these ORs and the PRS is inflated for some reason.

10. Given that coronary artery disease is a risk factor for AAA, did the authors investigate potential causality between the 2 conditions?

11. How do the authors justify the use of 0.5 as a threshold for significant colocalization (PP4)? This is quite liberal.

12. The PheWAS analysis and interpretation is not very well described. The authors did a look-up of their GWS hits in the IEU open GWAS project. Please describe how each snp was analysed against each phecode, did you apply any cut-offs for number of samples (total/cases), any cut-off for p-values based on multiple testing? The IEU project holds multiple copies of the same (or slightly different) dataset when outputting the PheWAS results for a query variant. Did the authors somehow curate the output before building the clusters? If duplicate association remained would that bias the clusters created? It would also be helpful for the reader to have a table version of the most significant PheWAS results with phecodes and summary stats.

Author Rebuttal to Initial comments

Reviewer 1

General comments: Roychowdhury et al. performed a meta-analysis of GWAS for abdominal aortic aneurysms using the largest sample size to date (39,221 cases), including European and African populations, and identified 97 novel loci and 144 independent signals. The PRS built on the summary statistics of this GWAS performed better than the one derived from the GWAS of Klarin et al. reported in 2020 (7,642 cases). The GWAS summary statistics and gene expression data also pinpointed genes, molecular pathways, and cell types that were prioritized for the disease. The genetic relationship with thoracic aortic aneurysms was also presented. Finally, using PheWAS, MR, and a mouse model, the authors proposed that PCSK9 inhibitors may be effective in the treatment of AAA. The downstream analysis is up-to-date and it is interesting to present the possibility of a new therapeutic agent. There are several suggestions and points of clarification I would like to make.

Major comments:

1. Discovery studies

Comment	In the studies used in the meta-analysis of the AAA discovery cohort (Figure S1 and Table S1), there were some studies with high or low lambda GC. Could you justify these inflated or deflated values? Especially, in order to interpret these values accurately (in particular, whether studies with high lambda GC were affected by polygenicity or not), LDSC intercept values may be helpful.
Response	We likewise appreciated the range of lambda GC values. We attribute this to polygenicity. As noted in table S1, we see highest GC lambda (MAF>0.05) for MVP (1.21, ncase=17672), followed by UKAGS (1.16, ncase=3595). These studies have the largest number of cases and thereby are most powered to capture polygenicity of AAA. To address this, we used a GC correction during meta-analysis. This step is used to correct potential inflation of test statistics in the discovery cohorts.
Excerpt from manuscript	Meta-analysis of 17 discovery cohort summary statistics was performed by METAL in standard error mode with genomic control correction.

2. Heritability

Comment	How much of the heritability can be explained by this AAA GWAS?
Response	We did not estimate liability-scale heritability of AAA as heritability calculations are highly dependent on the estimated population prevalence of the disease. For AAA, the population prevalence is difficult to estimate given the asymptomatic nature of AAA, geographic variance in screening, and the age dependent nature of the disease. Based on a pooled population prevalence of 4.8% (reported from a meta-analysis of 56 studies in PMID: 24312543), we estimate the liability-scale heritability of AAA = 0.1478 (0.0095). Using data from the individual studies comprising the meta-analysis, the heritability could vary widely (assuming a 1% prevalence reported individually in PMID: 20541348 or a 11% prevalence reported individually in PMID: 8564486). Given the wide range of values, we feel it is potentially misleading to present this data in the manuscript. We will be happy to add it if the reviewer feels strongly that it would enhance the message of the paper.

3. Replication

Comment	while I understood the usefulness of internal replication for novel loci, is external replication not possible? Finngen published summary statistics for AAA (https://r7.finngen.fi/pheno/19_ABAORTANEUR)
Response	We prioritized novel discovery over external replication. We have now performed a replication using the newly available Finngen data and the 2 cohorts we used for replication of the PRS that were not part of the genetic discovery analysis. These cohorts combined only have a total of 5,451 cases and are rather underpowered for replication. Moreover, several variants were absent in more than one external cohort (Table S2). Nonetheless, we observe a $p < 0.05$ and consistent direction of effects in 80/121 loci. Figure S7 demonstrates a consistency of effect estimates in well-powered loci. It is also evident from figure S7 that we primarily observe $P \geq 0.05$ in loci with weaker effect estimates, likely due to much smaller sample size in the replication cohort leading to lack of power.

	 Figure S7: Comparison of effect estimates (95% CI) in 121 loci between AAAGEN meta-analysis and external replication cohorts. The red dots represent 80 variants with $P < 0.05$ in external cohorts and the grey dots represent other 41 variants.
Excerpt from manuscript	We also observed consistent effect estimates (Figure S7) and $P < 0.05$ in 80/121 loci in external cohorts that were not included in the meta-analysis.

4. PRS

Comment	Main text, lines 203-212. Regarding the performance of PRS, just the values of Harrel's C-index were used to discuss whether the performance was better or not. However, you should test whether the difference was
--

	statistically significant using a method such as Bootstrap or something like that.
Response	As requested by the reviewer, we have now added optimism corrected C-indexes to the table S4b based on 100 times bootstrapping. C-index is not a model fit statistic but rather comparable to AUROC in the survival modeling and describes only the discriminative properties of the model. As it is not advised to evaluate models or their fit using P-values comparing two C-indexes, we have instead compared the model fit using likelihood ratio test and added the results to the table S4b.
Excerpt from manuscript	This model additionally showed superiority over the model with clinical factors only when comparing the model fit (P-value from the likelihood ratio test for nested models = 1.06E-101). We did not notice any notable optimism bias in the C-indexes when bootstrapping the statistics.

5. DEPICT analysis

Comment	Main text, line 214-. the DEPICT algorithm prioritized pathways, whereas the pathways available in the default setting were hard to interpret because there were so many similar ones. After clustering the pathways that exhibit similar functionality and obtaining the distinguished clusters (which include an 'exemplar' pathway and member pathways), could you perform the enrichment analysis using only the 'exemplar' pathways?
Response	We agree with reviewer about overlap and apparent redundancy in the “gene sets” employed by DEPICT. Unfortunately, this is a limitation of the available methodology. We used the “reconstituted gene sets” in DEPICT for this analysis. During the development of this software, authors used co-regulation data from 77,480 microarrays to predict genes’ biological functions across 14,461 gene sets representing a wide spectrum of biological annotations and to construct 14,461 “reconstituted gene sets”. In application, DEPICT identifies “reconstituted gene sets” that are enriched in the disease loci. The suggested change requires modification of the DEPICT algorithm itself and thus, we think out of scope for this manuscript.

6. Tissue/cell type enrichment

Comment	Main text, lines 219-234. The authors performed cell-type enrichment analysis using scRNA data from the aorta. On the other hand, in usual tissue enrichment analysis, which organs/ tissues are enriched with GWAS signals? What cell types are enriched in organs/ tissues associated with AAA other than the aorta?
Response	We employed tissue-specific chromatin marks to identify tissues that are enriched in AAA GWAS signals. As shown in figure 2b, the only tissue demonstrating significant enrichment was the aorta. This leads to our focus on the aorta for single cell RNA based analysis to identify enriched cell-types. We also showed expression of prioritized genes at the single cell level in figure S11.
Excerpt from manuscript	This analysis identified enrichment of per-SNP heritability in H3K27ac ($P=1.8\times 10^{-5}$), H3K4me1 ($P=2.6\times 10^{-5}$) and H3K4me3 ($P=7.8\times 10^{-5}$) marks in aorta (Figure 2b). Using regression coefficients, 4 out of 8 cell types were found to be enriched ($P<0.05/8$), with endothelial ($P=1.1\times 10^{-10}$) and smooth muscle ($P=1.4\times 10^{-9}$) cells being most strongly associated with AAA (Figure 2c). Most (87%) of the prioritized genes were expressed in the aorta, particularly in endothelial and smooth muscle cells, as observed in bulk and single cell RNA-seq of abdominal aorta from AAA patients (Figure S11)

7. Tissues for eQTL analysis

Comment	Main text, line 241. 4 tissues: aorta, liver, whole blood, and adipose. While I understand that the aorta was included, what criteria were used to select the other three?
Response	We used a few additional tissues as a subset of genes can be causal through tissues other than aorta. Our tissue-based rationale was as follows:

	 • Liver: relevant to lipid metabolism • Whole blood: aorta carries blood from heart to rest of the body • Adipose: relevant to storing excess lipids
--	--

8. Text clarification

Comment	Gene prioritization. Gene-prioritization strategies (main text, line 235-) identified putative causal genes at the AAA risk loci, whereas the integration with external datasets (main text, line 298-) elucidated AAA-associated genes in the aorta. This is confusing because both utilized TWAS, mouse AAA model, and human AAA gene expression data. Can you please sort this out?
Response	We have revised and reordered these sections to make things clearer. While utilizing some of the same approaches, these two sections serve separate purposes. The section on “Gene-prioritization identifies putative causal genes at AAA risk loci” outlines the gene prioritization approach. Gene expression markers (from previously obtained RNAseq data) and TWAS were a subset of 8 indicators we used in this step. Importantly, the starting point was GWAS itself with the goal of identifying the putative causal gene at each locus. The section “Integration with external datasets elucidates aortic expression of AAA associated genes” primarily focuses on direction of gene expression changes, using TWAS as a starting point, followed by data from mouse models of AAA. This section concludes with de novo qPCR validation of the intersection of the genes identified by TWAS and mouse models that were also prioritized in the previous section. We have changed the section header to read “Transcriptome-wide association study indicates the direction of gene-expression changes of AAA associated genes” to further clarify this.

9. Genetic correlation

Comment	Main text, lines 324-325. Showing the genetic correlations using GWAS summary statistics of AAA, TAA, and lipid traits could be useful to understand the context. I think you can use the European LD structure to calculate it since Europeans are the majority despite a multi-ancestry GWAS.
Response	As suggested by the reviewer, we have now used cross-trait LD-score regression to estimate the genetic correlations between AAA, TAA and lipid traits (Figure S13). * P < 0.0014 Figure S13: Genetic correlation (r_g) between thoracic/abdominal aortic aneurysm and lipid traits.
Excerpt from manuscript	Consistent with this, we observe a significant genetic correlation between lipids and AAA, but not TAA (Figure S13).

10. Genetic correlation

Comment	Main text, line368-370. The authors have checked major clinical risk factors for AAA using pairwise colocalization. Again, could you show us the genetic correlations R_g between these diseases ? Additionally, diabetes mellitus is known to be negatively associated with AAA or TAA (PMID: 30052821). How about the association between your AAA result and diabetes? Also, how about the association with other atherosclerotic cardiovascular diseases? (e.g. coronary artery disease)
Response	We appreciate the reviewer's interest in genetic correlations between cardiometabolic traits and AAA. The current colocalization analysis focuses on shared causal risk variants, while LD-score regression estimates shared polygenic architecture (and is thus not focused on individual risk variants). We have now performed LDSC to evaluate the genetic correlations between AAA and blood pressure, lipids, diabetes, smoking. We observe that AAA is genetically correlated with all common cardiometabolic traits (Figure S14). Of note, we see a positive correlation with T2D and are exploring the use of genetics to clarify the relationship between these traits as the focus of a separate project.

11. MR-BMA

Comment	Main text, lines 392-395. In MR-BMA analysis, instrumental variables were extracted from UK biobank-based lipid GWASs. Because your GWAS included UKBB samples, the assumption of two-sample MR that independent GWASs should be used for exposure and outcome was violated. Could you explain more about that?
Response	We appreciate the reviewer’s interest in the impact of sample-overlap on our MR-BMA findings. To our knowledge the impact of sample overlap in MR-BMA specifically has not been evaluated, however the impact in univariate two-sample MR more broadly has been previously explored

(e.g. PMID 27625185). The conventional advantage of two-sample MR with non-overlapping samples is that bias due to weak instruments is toward the null (that is no association between exposure and outcome), rather than toward the observational estimate as is the case with single-sample study designs. In the case of two-sample MR with a binary outcome, when associations between genetic variants and exposures are estimated in non-cases of the outcome of interest the exposure-outcome estimates remain unbiased. In the scenario where some cases are included in the studies used to estimate the associations between genetic variants and the exposure, effect estimates may theoretically become biased, with the degree of bias proportional to sample overlap but dependent on instrument strength. In this scenario, the degree of bias is proportional to sample overlap but dependent on instrument strength, such that strong genetic instruments derived from large consortia are less susceptible to bias due to sample overlap. In our MR analyses, genetic instruments for circulating lipids were constructed from a GWAS of 361,194 UK Biobank participants, while the corresponding effects of each variant on AAA were derived from our AAAgen study including 39,221 individuals with AAA and 1,086,107 controls. The maximum sample overlap between the populations is therefore estimated to be 32%. Using the calculator provided in PMID 27625185, we estimated the theoretical potential bias due to sample overlap. Due to the strong average instrument strength, bias due to sample overlap was determined to be negligible (while maintaining the type 1 error rate) for each of the lipid exposures (vertical red line denotes estimated proportion of sample overlap in the current study; the entire range of possible sample overlap is presented to demonstrate the negligible bias across all possible levels of sample overlap, across a range of potentially biased observational effect estimates):

These findings are consistent with the conclusion from PMID: 27625185

	that “Bias from weak instruments in very large consortia may not be substantial, but in moderately large consortia, potential bias, and inflated Type 1 error rates should be investigated.”
--	--

12. Mouse model

Comment	Main text, line 437-. Pcsk9^{-/-} mouse model. The authors mentioned that the PCSK9^{-/-} mouse model exhibited suppressed progression of AAA, in which how were the gene expression alterations of AAA in the mice? I think further molecular biological mechanisms can be identified by analyzing this point. Also, PCSK9 inhibition reduces lipid levels. To what extent were serum lipids reduced in the pcsk9^{-/-} mice? If the effect of PCSK9 inhibition was lipid-mediated, would lowering lipids by methods other than PCSK9 inhibition also strongly suppress AAA?
Response	We have performed gene expression analysis of samples derived from this study. As expected, expression of inflammatory cytokines IL-1b and Tnf-a were lower in Pcsk9^{-/-} animals compared to wildtype, while there were no appreciable differences in Lox-1, Acta2 or Bcl2 (see attached figure). These data indicate reduced vascular inflammation in Pcsk9^{-/-} animals. However, given the multiple mechanisms that interact throughout the development of AAA, more in-depth and temporal investigation is required, which we submit is beyond the scope of the current manuscript.

	Gene expression in AAA tissue 28d post PPE As has previously been reported (PMID 15805190), Pcsk9^{-/-} animals have reduced lipid levels compared to wildtype animals. Per the reviewer's request, we have now added the lipid values as a new Table S26. However, we and others have previously demonstrated that extreme elevation of cholesterol has no effect on the murine elastase AAA model (PMID 34680067; 32949946). In addition, pharmacological treatment with statins has demonstrated varying results upon elastase-induced AAA pathology as there are likely mechanisms beyond lipid lowering involved (PMID 16414398; 18482727; 23226500; 23711681). In summary, there is insufficient evidence to conclude that the modest changes in lipid values could account for the reduced experimental AAA development observed.
Excerpt from manuscript	As with previous studies⁵²⁻⁵⁴ Pcsk9^{-/-} mice had lower plasma cholesterol, LDL and HDL levels compared to wild type C57BL/6J mice (Table S26).

13. Discussion regarding PCSK9

Comment	PCSK9 inhibitors are one of the options for lipid-lowering therapy in the presence of atherosclerotic diseases. It would be interesting to explain a bit more about the relationship between AAA and other atherosclerotic diseases via PCSK9.
---

Response	We agree that these findings suggest a broader role for other lipid-lowering therapies in AAA and have added a line in the discussion to reflect this.
Excerpt from manuscript	Here, we expand upon these findings by observing that 42 of the 121 AAA risk loci are also associated with lipids, supporting the notion of AAA as an end-organ manifestation of atherosclerosis and that lipid modulating therapies may have a role in the management of AAA. We additionally prioritize lipid subfractions beyond LDL-C, namely remnant cholesterol, as likely causally related to AAA. Molecules targeting the lipoprotein lipase pathway via APOC3 are emerging therapies for cardiovascular disease but remain unstudied in aneurysm patients. More broadly, our findings suggest that AAA shares important pathobiology with other forms of atherosclerotic vascular disease and should be considered on this disease spectrum.

Minor comments:

1.

Comment	GO analysis (Table S15). GO proposes molecular functions, biological processes, and cellular components. Where is the biological process?
Response	In response to the reviewers query we have now added this to Table S15.

2.

Comment	Manhattan Plot (FigureS3. It will be easier to understand if you can plot the new loci in a different color. Please also provide the name of the novel loci.
Response	We have modified this accordingly.

3.

Comment	Figure S4, S5. If the range of 95% confidence interval is shown, we can understand how accurate the effect estimates were.
Response	We have added 95% confidence interval in these plots.

4.

Comment	“FDR P-value” is an odd expression and should simply be “FDR” or “q-value”.
Response	We modified the language to “FDR” throughout the manuscript and supplemental material.

5.

Comment	Main text, line 1073. The expression “Welch's t-test for parametric and Mann-Whitney for non-parametric”. This is an odd expression. The t-test is a parametric test and assumes a normal distribution, while the Mann-Whitney test is a nonparametric test and is used when a normal distribution cannot be assumed.
Response	For the mouse experiments we used the Mann-Whitney test. We have changed this accordingly throughout the manuscript.

Reviewer 2

General comments: This paper describes the results of the largest GWAS to date for AAA – with 97 new loci. The authors have performed comprehensive downstream analyses with the new findings providing new insights into underlying disease biology, and demonstrate overlap with other aortopathies. The PRS work can explain more than the clinical risk factors and they highlight results to indicate that PCSK9 is a therapeutic target.

The authors have done a tremendous amount of work, and the results contribute new knowledge to the causes of AAA with discovery of new loci for this condition and highlight some therapeutic options for consideration.

Major comments:

1. Rare variant exclusion

Comment	The description of the meta-analysis was clear to follow and nice to see the internal validation you did. You indicate removal of rare variants, from looking at the individual studies it does not look like a MAF threshold was used? Therefore, it was a little odd to take variants all the way forward and then remove, as the variant passed internal validation – or did it, this is not clear from text or suppl. Table 2.
Response	Our apologies for the confusion. The two variants were not excluded based on MAF threshold (0.01), although they were rare variants. Rather, after the meta-analysis, two loci that passed internal replication or sensitivity analysis were excluded due to following reasons: i) the index variant was the only genome-wide significant variant at the locus and not supported by any additional variants; ii) the index variant was only present in the two MVP specific GWAS and no other cohorts. Thus, we lacked confidence in these two variants being true positives. We have clarified this in text.
Excerpt from manuscript	Additionally, for 2 loci we lacked confidence, as genome-wide significant rare index variants were not supported by any additional variants in these loci.

2. PRS text

Comment	You performed several PRS analysis – the motivation for all the different analyses was not really described – you would have predicted your new PRS would outperform the Klarin one, I thought the text in this section could be reduced.
Response	We appreciate the reviewer’s comments. We have clarified our motivation

	for our PRS analyses. However, we added some additional text in this section following comments from reviewers 1 and 3.
Excerpt from manuscript	These two sets of weights were then used to calculate PRS in three external validation cohorts, two with control groups representative of the general population (UKAGS-UKB and AC-WTCCC) and one representative of populations with cardiovascular disease (SMART). AAAgen PRS performed less well in the SMART cohort, in which the controls also had prevalent (non-aneurysmal) cardiovascular disease, but still outperformed the Klarin et al. PRS (AUC 0.64 vs 0.59, $P < 0.0001$). This also indicates a utility of PRS in distinguishing aneurysmal risk from broad cardiovascular risks for clinical applications (Figure 1c, Figure S8, Table S4a).

3. Gene prioritization text

Comment	The gene prioritisation section I found difficult to follow and long. I appreciate all the analyses you did but it is not clear at the end which genes have strong evidence and from which sources. I appreciate creating a way to prioritise candidate genes is quite subjective and across GWAS studies there is no wrong or right approach. What you have done however does need to be clearer. I wondered if you can provide an overview figure of what was done here for the reader? At the outset you do describe 8 different indicators, so at the outset what was the starting number of candidate genes? You move to then describing 523 genes with 5 of the above indicators, so there are 523 genes which have support from 5/8 indicators. I got more confused when you started with describing results following rules again in the second last paragraph starting with “next we used the following rules”. There is no page numbers for me to reference. I think the whole section on gene prioritization needs a rethink in reporting so it is clear what the input was for testing for enrichment with GO. Once this section is more clearly described, then highlight as you have done interesting pathways and genes.
Response	We apologize for the confusing nature of our description of the analysis. We have completely rewritten this section of the manuscript in a way that we hope is clearer and, as suggested by the reviewer, added a flowchart

4. Pairwise GWAS

Comment	I wondered for formally assessing the overlap with thoracic aortic aneurysm if you should consider using pairwise- GWAS (https://github.com/joepickrell/gwas-pw)?
Response	We appreciate the suggestion of the reviewer as we are keenly interested in the relationship between TAA and AAA. Given our current GWAS findings for AAA and our recent work on TAA (https://doi.org/10.21203/rs.3.rs-1507463/v1) we are engaged in a large scale effort to compare the pathobiology of these seemingly related entities using approaches similar to what the reviewer suggests. We feel that a full treatment of this comparison is beyond the scope of the current paper.

5. Order of sections

Comment	I thought once the gene prioritisation section was clearer, the section labelled “integration with external datasets elucidates...” Could follow this, the overlap with other traits follows this section not precedes as it currently stands.
Response	We thank the reviewer for this suggestion. We have now changed the order of these sections.

6. Blood pressure

Comment	In the section “pleiotropic AAA risk...” I did wonder if you also wished to state a little more on the relationship with BP as well?
Response	We appreciate the reviewer’s interest in exploring the relationship between blood pressure and AAA in more detail. As previous studies (Klarin et al.) by our group have robustly explored this relationship using Mendelian randomization, we do not feel that there is novelty in adding similar analysis. Rather, a novel finding from our current analysis is the central role of lipoprotein biology in AAA, which is what we have chosen to focus on in the current manuscript.

7. MR in abstract

Comment	The MR section and leveraging potential drug purposing opportunities I thought was interesting and the results from the mouse model. I wondered if this work can be highlighted a little more in the abstract. You provide lots of text on genes and pathways, these are results of equivalent interest to readers with the availability of PCSK9 inhibitors.
Response	Thank you for your interest in this analysis. We have modified the abstract accordingly.
Excerpt from manuscript	Motivated by the strong evidence for the role of lipid metabolism in AAA, we utilized Mendelian Randomization to establish the central role of non-high density lipoprotein cholesterol in AAA and identified the opportunity for

	repurposing of PCSK9 inhibitors. This was supported by a study demonstrating PCSK9 loss of function prevented the development of AAA in a pre-clinical mouse model.
--	--

Minor comments:

1.

Comment	Add a Table of Contents for the Suppl. Tables.
Response	We have added this.

2.

Comment	Figure 3 is great and summarises the results nicely, I would expand in the main text describing this figure when you reconfigure reporting of your gene prioritisation pipeline.
Response	Thanks for the suggestion. Figure 3a summarizes the support of various indicators for the prioritized genes. In addition to reconfiguring the text, we have now highlighted this.

3.

Comment	Figure 5 – figures and text needs to be bigger on axes
Response	We have modified this.

4.

Comment	Figure 7 – can you expand the legend to indicate source of your categories for the genes from the suppl. Tables.
Response	We have added this.

5.

Comment	Suppl Table 14 is great, can you provide some text as a legend to explain the final gene selection in column AC
Response	We added this information.

Reviewer 3

General comments: Roychowdhury et al. have performed a GWAS meta-analysis in AAA and followed up the findings with downstream functional and expression analysis. While the results are impressive there are some major concerns regarding the methods and interpretation of the results.

Major comments:

1. Use of the term “Multi-ancestry”

Comment	I found the use of the term “multi-ancestry” in the title and abstract quite misleading. Apart from the very small fraction of African Americans included in the meta-analysis, there is no other ancestries included, no ancestry-specific analysis, no estimation of the gains in PRS performance or fine-mapping improvement from the addition of non-European ancestries. I do not consider this study as multi-ancestry in that respect. All downstream analysis are also limited to European reference panels or annotation features.
----------------	---

Response	We have excluded the term multi-ancestry from the manuscript. The new title is “GWAS meta-analysis deciphers genetic architecture of abdominal aortic aneurysm and highlights PCSK9 as a therapeutic target”.
-----------------	--

2. Discovery studies

Comment	It is not very clear whether this meta-analysis was based on a uniform analysis plan for all contributing studies. There is a range of different covariates used from the studies that do not necessary fall under the cohort-specific population stratification adjustments. For example ARIC is adjusting for smoking, DiscovEHR for a series of age and gender combinations, eMERGE only for PCS, there is no mention of covariates for NZ AAA Genetics studies etc. It would be helpful to supply a supplementary table with cohort-specific analytical details. Do the authors fear that this lack of standardisation in the analysis might have impacted the results?
Response	We appreciate the reviewer’s interest in understanding the covariates used in the primary analyses. As there were large differences in the cohorts, and many cohorts had existing AAA analyses that were subsequently contributed to this effort, we gave significant leeway to the cohorts with respect to primary analysis of their data. Admittedly, this could have been more clearly presented. As per the reviewer’s suggestion, we have added the covariates used by each cohort to table S1. To address potential concerns regarding the effect of non-standard primary analyses, we have now performed “leave one cohort out” meta-analyses without cohorts where covariates were different from majority of the discovery studies (Figure S6). Performed at all genome-wide significant loci, these analyses show that there are no clear outliers (variants driven by the adjustment scheme) or trends (analysis biased because of the adjustment scheme).

	Figure S6: Comparison of effect estimates (95% CI) for 121 index variants between AAgen meta-analysis and leave-one-cohort-out meta-analysis for six discovery cohorts.
Excerpt from manuscript	We performed additional sensitivity analysis for remaining 121 loci and observed consistent effect estimates across cohorts (Figure S6).

3. UK biobank

Comment	The use of UK Biobank in this meta-analysis and subsequent analysis is very difficult to follow. I suggest the authors provide with a diagram for that. My main questions are: 1. Why are there only 6,276 controls in the discovery stage and how were these selected? 2. When testing the PRS in the UKAGS-UKB validation study it is not clear if the PRS is with or without UK Biobank, 3. Was the MR performed in the UK Biobank with the meta-analysis summery statistics with or without UK Biobank, 4. Please clarify the adjustment used for the association analysis and whether relatedness was considered, 5. Please clearly mention in the main text the number of incident AAA cases in UK Biobank.
Response	We apologize for the confusion relating to our use of UKB data. Prior to any analyses it was divided into 3 non-overlapping subsets to generate the following:

1. Control data for UKAGS cases used in Discovery
2. Control data for UKAGS cases used in Validation
3. An analysis of UKB participants with AAA

With respect to the reviewer's specific questions on the use of UKB data:

1. The aim of the UKB-UKB GWAS was to have a 1:5 case:control ratio. It is approximately this, i.e., 1241:6276. It is not a precise 1:5, due to exclusion of samples in the QC procedures. The same approach was used for the UKAGS-UKB discovery GWAS (3595:15773) and UKAGS-UKB validation cohort (1130:5810).

2. Testing the PRS in the UKAGS-UKB validation study (Figure 1C) was performed with full meta-analysis of AAAGEN. There is no sample overlap between meta-analysis and testing as separate UKAGS cases and UKB controls were used for these (figure S17).

3. MR-BMA analysis was performed with the meta-analysis including UK biobank. We answered this question in detail in response to reviewer 1 (comment 11). In summary using additional analysis, we show that bias generated in MR analysis by sample overlap is negligible for meta-analysis with large sample size.

4. Related individuals were excluded, by removing 1 of any pairwise kinships.

5. We used 838 incident cases in UK biobank and added this information in the text.

We added figure S17 to clarify the usage of UK biobank controls.

Usage of UK Biobank controls					
	   Discovery cohorts Validation cohorts      UKBB cases, n = 1241 UKBB controls, n = 6276 UKAGS (cluster 1 and 2) cases*, n = 3595 UKBB controls, n = 15773   UKAGS (cluster 3) cases**, n = 1130 UKBB controls, n = 5810     * includes 386 Danish (VIVA) samples ** includes 71 Oxford (OxAAA), 247 Swedish (UppsalaAAA) and 4 Danish (VIVA) samples Figure S17: Diagram for the usage of UK Biobank controls.	Discovery cohorts	Validation cohorts	UKBB cases, n = 1241 UKBB controls, n = 6276 UKAGS (cluster 1 and 2) cases*, n = 3595 UKBB controls, n = 15773	UKAGS (cluster 3) cases**, n = 1130 UKBB controls, n = 5810
Discovery cohorts	Validation cohorts				
UKBB cases, n = 1241 UKBB controls, n = 6276 UKAGS (cluster 1 and 2) cases*, n = 3595 UKBB controls, n = 15773	UKAGS (cluster 3) cases**, n = 1130 UKBB controls, n = 5810				
Excerpt from manuscript	UKBB subjects were included for a UKBB-UKBB case-control association study, and to provide control subjects for each of the separate case groups used in the discovery and validation cohorts (see Additional validation cohorts for PRS), matching for sex and age, and without overlap of subjects between analyses. See Figure S17 for the usage of UK biobank controls in various analyses. Individuals included from both studies were of white British/Danish ancestry, with PCA identified outliers excluded; related individuals were identified with PLINK v1.9 IBD computation (--genome), and one of any pairwise kinships removed. The final analysis included 838 incident AAA cases and 329,983 non-cases with median follow-up time of 5.04 and 10.0, respectively.				

4. Case number in MVP

Comment	The MVP study seems to have disproportionately higher number of AAA cases compared to Biobanks of similar size (eMERGE or UK Biobank). How do the authors explain that? Could that be due to misclassification?
Response	We believe this is primarily due to differences in cohort recruitment.

	Million Veteran Program recruits Veterans seeking health care in the US Veterans Administration Healthcare System. This population is primarily composed of older males with a high rate of prior or current use of smoking tobacco and would thus be expected to be enriched in AAA. The MVP phenotyping algorithm was previously validated by manual chart review (PMID 30090940).
--	---

5. Replication

Comment	The authors perform the meta-analysis and then use the same meta-analysis without the MVP as a replication step. This might be considered a sensitivity analysis but not replication. The authors should use the PRS validation studies for replicating their GWS findings.
Response	As suggested by the reviewer, we changed the terminology to sensitivity analysis. We have now performed a replication using the newly available Finngen data and the 2 cohorts we used for replication of the PRS that were not part of the genetic discovery analysis. These cohorts combined only have a total of 5,451 cases and are rather underpowered for replication. Moreover, several variants were absent in more than one external cohort (Table S2). Nonetheless, we observe a $p < 0.05$ and consistent direction of effects in 80/121 loci. Figure S7 demonstrates a consistency of effect estimates in well-powered loci. It is also evident from figure S7 that we primarily observe $P \geq 0.05$ in loci with weaker effect estimates, likely due to much smaller sample size in the replication cohort leading to lack of power.

	 Figure S7: Comparison of effect estimates (95% CI) in 121 loci between AAAGEN meta-analysis and external replication cohorts. The red dots represent 80 variants with $P < 0.05$ in external cohorts and the grey dots represent other 41 variants.
Excerpt from manuscript	Since approximately 45% of all cases were contributed by the VA Million Veteran Program (MVP) EUR analysis, we performed another meta-analysis without this cohort and tested for nominal significance ($P < 0.05$) in both datasets as sensitivity analysis. We also observed consistent effect estimates (Figure S7) and $P < 0.05$ in 80/121 loci in external cohorts that were not included in the meta-analysis.

6. Joint analysis

Comment	The authors claim 121 independent loci but these are only identified by distance pruning, therefore not tested for independence. The authors should apply the cojo-joint in GCTA to assess independence. On a similar note, it is not clear why the authors use the cojo-cond for identifying the additional 24 secondary loci, as the cojo-joint is a better approximation of the exact conditional analysis. Please also provide joint p-values in the supplementary table.
Response	As per reviewer's request, we have now performed a joint-analysis using an LD panel generated from a subset of PMBB participants that had the same proportions of EUR and AFR ancestry participants as the overall meta-analysis. Using this LD panel, we are unable to obtain the joint p value of 4/121 index variants since these 4 variants are missing in PMBB. We added this information in table S2. We also used cojo-joint to calculate p-values for the secondary signals at each locus and have added this information to table S3. Following joint analysis, we excluded 3 variants from this set.
Excerpt from manuscript	Definition of loci: Independent loci were defined as variants >1 Mb and >0.25 cM apart with at least one genetic variant associated with AAA at a genome-wide significance threshold of $P < 5 \times 10^{-8}$. Index variants are the variants with lowest association p-value in every locus. Index variants were tested for independence using GCTA COJO (cojo-joint) as described below. Conditional analysis: To perform conditional analysis, we first defined the loci as +/- 1 Mb from each of the 121 index SNPs. GCTA COJO and specifically the cojo-cond function was performed iteratively using a reference panel created from individuals in Penn Medicine BioBank that represented the demographics of AAAGEN (17:1 EUR to AFR ancestry). Iterations were performed at each locus until a minimum number of independently, genome-wide significant SNPs were identified. The maximum number of iterations performed was 4 at the rs10455872 locus. Next, we performed a joint analysis of variants using cojo-joint with the same reference panel and excluded variants that were not genome-wide significant in joint analysis.

7. Cox model

Comment	Please add the covariates used in the cox model methods section.
Response	The full models are described in the table S4b. We have now added a clear reference to this table in the Methods section. The possible population stratification effects of the PRS were accounted for by adjusting the PRS with genetic principal components before the survival analyses and hence the PCs were not included in the cox-models.
Excerpt from manuscript	All variables used in the different models are listed in the Table S4b.

8. PRS computation

Comment	The PRS computation is not clearly described. Please mention which tuning criteria were used and how you chose the best performing PRS. Did you test it in an independent study?
Response	We used PRScs with default parameters to calculate the PRS, which obviates the need for a tuning step. For a polygenic trait GWAS with large sample size, PRScs can efficiently generate the global shrinkage parameter phi using a Bayesian approach from the input summary data. We clarified this in the text.
Excerpt from manuscript	We used default parameters of PRScs, allowing the method to generate the global shrinkage parameter phi through a Bayesian approach.

9. Odds ratio from PRS

Comment	Figure S6. The ORs for the current PRS are unrealistic and so are the CIs, especially for the right-hand side deciles. It seems unlikely that the validation studies have the statistical power to detect these ORs and the PRS is inflated for some reason.
Response	AAA diagnosis is based on accurate and simple imaging so phenotypic definition of cases is extremely robust. The low population prevalence of

AAA (less than 5%) means that control cohorts are relatively uncontaminated with occult cases. In datasets such as UK Biobank (UKB), used here for the comparison of the additional UK Aneurysm Growth Study (UKAGS) samples, the UK aneurysm screening programmes for men will have reduced the number of occult AAA in the control group further. This results in very well characterized case and control groups and may explain the ORs observed in this research, particularly in the UKAGS-UKB analysis. In addition, it is important to note the relatively high heritability of AAA when compared to other complex traits where PRS have been examined, which may explain these findings.

Despite this potential explanation, we acknowledge that the ORs for the new PRS are high, particularly for the highest decile.

To address this, we have re-analyzed our datasets to check for errors. The only error we identified was that PRS decile groups had been defined using the PRS for all individuals in combination rather than for each cohort separately. We have corrected this error and the revised data is presented. This had no effect on the interpretation of these findings, with the AAAgen PRS demonstrating an improvement in performance compared to the MVP PRS (Klarin et al.) as would be expected for this much larger study, and in particular with the most robust phenotypic definition for AAA, the UK Aneurysm Growth Study.

Regarding the high ORs presented, we have deliberately not emphasized this finding in our manuscript and the confidence intervals demonstrate the uncertainty around these estimates. We do consider the presentation to be an accurate representation of our findings. To improve the clarity and interpretability however, we have added the case and control numbers in each decile as a sub-table within our revised figure S6, now figure S8.

To further address this comment, we have re-analyzed our original two validation datasets using the combined 5th and 6th deciles as the reference category. This analysis is presented in comparison to the original data in the figures below:

Original analysis with first decile as reference:

As can be seen the re-analysis reduces the OR estimates in each decile but does not alter the relative difference in ORs between deciles.

Our chosen approach in our revision is to use our original approach but to supplement this with the count data for cases and controls in each decile.

Prompted by this comment, and in addition to addressing the high observed ORs in the validation cohorts, we decided to explore PRS performance further by examining a cohort that may be more representative of populations with prevalent cardiovascular disease. We obtained data from the SMART study (Utrecht). SMART is a cohort study that has been ongoing since 1996. The SMART cohort consists of people presenting with cardiovascular disease at the University Medical Centre, Utrecht. We undertook a GWAS of SMART, comparing those with AAA to those with other cardiovascular disease. The AAAgen PRS performed less well in this cohort compared to the population-based cohorts but still demonstrated reasonable performance, and an improvement on the Klarin et al. PRS.

As a further check we have examined PRS score density plots for each score/cohort, which provided further reassurance that our results are accurate:

Excerpt from manuscript

AAAGen PRS performed less well in the SMART cohort, in which the controls also had prevalent (non-aneurysmal) cardiovascular disease, but still outperformed the Klarin et al. PRS (AUC 0.64 vs 0.59, P<0.0001). This also indicates a utility of PRS in distinguishing aneurysmal risk from broad cardiovascular risks for clinical applications (Figure 1c, Figure S8, Table S4a).

10. CAD

Comment	Given that coronary artery disease is a risk factor for AAA, did the authors investigate potential causality between the 2 conditions?
Response	In this paper, we investigated known risk factors of AAA, i.e., lipid levels, smoking, and BP. Exposure to these risk factors increase an individual's chance of developing AAA. By that definition, we can't consider CAD as a risk factor, but another disease with overlapping disease etiology. We plan to investigate this overlap in a future work.

11. Colocalization

Comment	How do the authors justify the use of 0.5 as a threshold for significant colocalization (PP4)? This is quite liberal.
Response	We agree to the reviewer that 0.5 is a liberal threshold and changed the threshold to 0.8 for trait colocalization analysis. In supplementary table (Table S7), we listed genes that pass 0.8 threshold for eQTL colocalization. However, we continued to use the 0.5 threshold for gene prioritization for the following reasons: i) This evidence was used at stage 1 to identify candidate genes and we looked for additional gene expression evidence from human/mouse AAA for these candidate genes at stage 2. A liberal threshold is better for not missing out candidate genes with evidence of differential gene expression at stage 2. ii) We also used TWAS methodology to collect evidence of eQTLs iii) Evidence of eQTL was one of the eight indicators used for gene prioritization and no gene was prioritized based on eQTL evidence only. Thus, we feel a liberal threshold is better suited for this scenario.
Excerpt from manuscript	At a threshold of $PP4 > 0.8$, we observed colocalization of 26 AAA loci with lipids (including PCSK9, $PP4=1$ for both LDL and total cholesterol), 9 loci with blood pressure (Table S20).

	A threshold of $PP4 > 0.5$ was used as evidence of colocalization for identifying candidate genes at stage 1 of gene prioritization. This analysis identified 82 loci with at least one gene (Table S7). In table S7, we also listed genes with stronger colocalization evidence ($PP4 > 0.8$).
--	---

12. PheWAS

Comment	The PheWAS analysis and interpretation is not very well described. The authors did a look-up of their GWS hits in the IEU open GWAS project. Please describe how each snp was analysed against each phecode, did you apply any cut-offs for number of samples (total/cases), any cut-off for p-values based on multiple testing? The IEU project holds multiple copies of the same (or slightly different) dataset when outputting the PheWAS results for a query variant. Did the authors somehow curate the output before building the clusters? If duplicate association remained would that bias the clusters created? It would also be helpful for the reader to have a table version of the most significant PheWAS results with phecodes and summary stats.
Response	We added additional text to clarify these questions (see below). As requested by the reviewer, we also added a table version of significant PheWAS results in table S18.
Excerpt from manuscript	Based on follow-up network analysis (see Methods) using PheWAS summary statistics, we identified 7 distinct modules of phenotype-clusters (Figure 4a, Table S18-19). Furthermore, the full summary statistics were filtered to GWAS studies that contained 50 or more cases. When the trait was measured at multiple locations (i.e., right and left arm) only results from one side were queried. For replicate phenotypes, the largest study of that phenotype was chosen. PheWAS results for each independent variants were considered significant if they passed a Bonferroni corrected threshold for 161,535 (121 variants and 1335 phenotypes) tests conducted.

Decision Letter, first revision:

3rd Apr 2023

Dear Dr Damrauer,

Your Article, "GWAS meta-analysis deciphers genetic architecture of abdominal aortic aneurysm and highlights PCSK9 as a therapeutic target" has now been seen by 3 referees. You will see from their comments below that while they find your work of interest, some important points are raised. We are interested in the possibility of publishing your study in Nature Genetics, but would like to consider your response to these concerns in the form of a revised manuscript before we make a final decision on publication.

We therefore invite you to revise your manuscript taking into account all reviewer and editor comments. Please highlight all changes in the manuscript text file. At this stage we will need you to upload a copy of the manuscript in MS Word .docx or similar editable format.

*2) If you have not done so already please begin to revise your manuscript so that it conforms to our Article format instructions, available [here](http://www.nature.com/ng/authors/article_types/index.html). Refer also to any guidelines provided in this letter.

[redacted]

We hope to receive your revised manuscript within four to eight weeks. If you cannot send it within this time, please let us know.

Sincerely,
Wei

Wei Li, PhD
Senior Editor
Nature Genetics
New York, NY 10004, USA
www.nature.com/ng

Reviewers' Comments:

Reviewer #1:

Remarks to the Author:

I am pleased with the effort to integrate so many heterogeneous studies as robustly as is currently possible and also with the responses that make it very clear which parts of the manuscript have been updated. I also have no additional comments other than one comment and one minor question.

1. Regarding the "exemplar" pathways in DEPICT, you can refer to the paper:
<https://www.nature.com/articles/s41588-020-0705-3#Sec15>

2. What is the reason for avoiding a direct comparison of AAA with coronary artery disease as suggested by Reviewers #1 and #3?

Reviewer #2:

Remarks to the Author:

Thank you for addressing each of the points I raised, most of these you have addressed satisfactorily however I have a few outstanding queries relating to gene prioritisation remaining which I summarise below.

Section: GWAS meta-analysis identifies 97 novel risk loci.

In your revised GWAS reporting can you indicate what the external cohorts were, numbers, ancestry and the motivation for this analysis. These are new analyses that were added in revision so some context on why you did this is now required.

Section: Gene prioritisation identifies putative causal genes at AAA risk loci.

The text in this section is now a little clearer on the process followed and the overview figure helps. Regarding the overview figure I think this can be improved and it should be a main figure. You should add information to the legend to ensure further clarity and further information to the figure. With 121 lead variants – indicate this is the number of loci being considered for stage 1 and your goal to identify just one gene per locus. In the legend indicate any one of these criteria was used for identifying a candidate gene at a locus. Stage 2 indicates 523 genes at 141 loci right, so in legend indicate additional lines of evidence was reviewed for each gene as more than one gene at some loci. Would the last box then be a stage 3, using all 8 indicators you prioritised candidate genes firstly on protein altering variants (are these predicted to be damaging by informatics tools...), support from a least indicators in stages 1 and 2 and finally if no single candidate gene indicated at a locus then you picked the nearest gene for that locus. I am summarising my understanding of what you have done in the above text, but hope in doing this if correct this process can be clear in your figure and reporting. I note there are genes at 37 of the loci not having support from 7 of your criteria but genes at these loci were included for testing for enrichment in GO terms. Noting your comment in the text that 87% were expressed in the aorta (were the 37 genes?), this is worth commenting on. As you brought forward all the genes at the 121 loci for enrichment testing, it would be interesting to see your results without the 37 genes based on distance only if not expressed in aorta. I noted that for the secondary signals you included genes only based on distance why was the same process not followed for selection of candidate genes as the lead signal?

Transcriptome-wide association study indicates the direction of gene-expression changes of AAA associated genes

You performed a TWAS in aortic tissue leveraging 104 genes these were then explored looking at differential expression in mouse and human tissues. The outcome from this analysis was 23 genes with consistent directions of effect and only 11 of these were prioritised as causal genes from your earlier pipeline. Subsequent work validated the 11 genes. I think you need to include your motivation for the TWAS as a method for prioritising candidate genes as a separate analysis to the pipeline you followed above – why this was done needs to be clearer as others using this tool as a way to prioritise genes and you could have used the results as an indicator. The results differ – with little overlap,

validation of genes from TWAS used same datasets as your earlier pipeline. I understand you wish to highlight the most promising candidate genes as best you can with the existing data but some discussion on the motivation for each of your analyses is required and why results are discrepant, now only 11 genes and these are the ones you focus discussion on. Can you map back the 11 genes as having eQTLs from stage 1?

Reviewer #3:

Remarks to the Author:

The authors have sufficiently addressed all my comments and have amended the manuscript accordingly.

Author Rebuttal, first revision:

Reviewer 1

General comments: I am pleased with the effort to integrate so many heterogeneous studies as robustly as is currently possible and also with the responses that make it very clear which parts of the manuscript have been updated. I also have no additional comments other than one comment and one minor question.

14. DEPICT analysis

Comment	Regarding the "exemplar" pathways in DEPICT, you can refer to the paper: https://www.nature.com/articles/s41588-020-0705-3#Sec15
Response	We thank the reviewer for pointing out the relevant resources. Accordingly, we identified representative "exemplar" gene-sets from DEPICT analysis and updated Figure 2a.
Excerpt from manuscript	The representative gene-sets indicated diverse biological functions as blood vessel development, hemorrhage, abnormal liver morphology, decreased erythrocyte cell number, increased inflammatory response etc. (Figure 2a).

15. Coronary artery disease

Comment	What is the reason for avoiding a direct comparison of AAA with coronary artery disease as suggested by Reviewers #1 and #3?																									
Response	We investigated the association of putative causal risk factors of AAA, i.e., lipid levels, smoking, and BP. Exposure to these risk factors increase an individual's chance of developing AAA. As we don't consider coronary artery disease a risk factor, but another disease with overlapping disease etiology, we did not originally investigate it. We believe, a thorough investigation of this overlap requires further work that is out of scope of this manuscript. As per reviewer's request, however, we have now added genetic correlation with coronary artery disease, peripheral artery disease and large-artery stroke.   <caption>Data for Figure S15: Genetic correlation (r_g) between AAA and cardiovascular diseases</caption>    AAA CAD PAD LAS     LAS 0.4 0.6* 0.8* 1.0   PAD 0.6* 0.6* 0.8 0.8*   CAD 0.6* 0.8 0.6* 0.6*   AAA 1.0 0.6* 0.6* 0.4    * $P < 0.0083$  Figure S15: Genetic correlation (r_g) between AAA and cardiovascular diseases (CAD: coronary artery disease; PAD: peripheral artery disease; LAS: large artery stroke).		AAA	CAD	PAD	LAS	LAS	0.4	0.6*	0.8*	1.0	PAD	0.6*	0.6*	0.8	0.8*	CAD	0.6*	0.8	0.6*	0.6*	AAA	1.0	0.6*	0.6*	0.4
	AAA	CAD	PAD	LAS																						
LAS	0.4	0.6*	0.8*	1.0																						
PAD	0.6*	0.6*	0.8	0.8*																						
CAD	0.6*	0.8	0.6*	0.6*																						
AAA	1.0	0.6*	0.6*	0.4																						
Excerpt from	We observed significant genetic correlation between AAA and common cardiometabolic risk factors (Figure S14) as well as other vascular diseases,																									

manuscript	including coronary artery disease (Figure S15).
------------	--

Reviewer 2

General comments: Thank you for addressing each of the points I raised, most of these you have addressed satisfactorily however I have a few outstanding queries relating to gene prioritisation remaining which I summarise below.

8. Replication analysis

Comment	Section: GWAS meta-analysis identifies 97 novel risk loci. In your revised GWAS reporting can you indicate what the external cohorts were, numbers, ancestry and the motivation for this analysis. These are new analyses that were added in revision so some context on why you did this is now required.
Response	We incorporated the replication analysis in the last round at the request of reviewer 1 and 3. In general, the scope of replication for such large meta-analysis is limited due to lack of external cohorts and thereby lack of power. However, using small amount of external data, we were able to demonstrate consistent effect size estimation. We added the description, case number, ancestry etc. to the results section (see below) and Table S2. Further details of these cohorts are described in methods section describing “external replication cohorts”.
Excerpt from manuscript	We also observed consistent effect estimates (Figure S7, Table S2) and $P < 0.05$ in 80/121 loci in external replication cohorts (total 5,451 cases of EUR ancestry from FinnGen and PRS validation cohorts).

9. Gene prioritization

Comment	Section: Gene prioritisation identifies putative causal genes at AAA risk loci. The text in this section is now a little clearer on the process followed and the overview figure helps. Regarding the overview figure I think this can be improved and it should be a main figure. You should add information to the legend to ensure further clarity and further information to the figure. With 121 lead variants – indicate this is the number of loci being considered for stage 1 and your goal to identify just one gene per locus. In the legend indicate any one of these criteria was used for identifying a candidate gene at a locus. Stage 2 indicates 523 genes at 141 loci right, so in legend indicate additional lines of evidence was reviewed for each gene as more than one gene at some loci. Would the last box then be a stage 3, using all 8 indicators you prioritised candidate genes firstly on protein altering variants (are these predicted to be damaging by informatics tools...), support from a least indicators in stages 1 and 2 and finally if no single candidate gene indicated at a locus then you picked the nearest gene for that locus. I am summarising my understanding of what you have done in the above text, but hope in doing this if correct this process can be clear in your figure and reporting.
Response	As per the reviewer’s suggestion, we added the gene-prioritization flowchart as main figure (Figure 3a) and added an explanatory legend.

Figure 3: Gene prioritization. a) Flowchart for the gene-prioritization pipeline. The primary goal of this pipeline was to identify a single gene for each of the 121 genome-wide significant loci. Evidence of 8 indicators were collected in two stages so that each prioritized gene was supported by at least one indicator from stage 1. At stage 1, we used 5 indicators to collect evidence for all genes within 1 mb of index variants. This procedure identified 523 candidate genes with evidence of at least 1 indicator. At stage 2, evidence of three additional indicators were collected for these 523 genes. Finally at stage 3, above 8 indicators were combined in three steps in order of precedence to identify single prioritized gene for 121 loci. **b)** Support of various gene prioritization indicators for 84 loci where a putative causal gene could be prioritized by protein-altering variants or by consensus. Rows (gene names)

	represent these loci and columns represent 8 supporting indicators used for the prioritization. Black dots at row/column intersection indicate support by a particular indicator for a particular gene.
--	---

10. Gene prioritization

Comment	Section: Gene prioritisation identifies putative causal genes at AAA risk loci. I note there are genes at 37 of the loci not having support from 7 of your criteria but genes at these loci were included for testing for enrichment in GO terms. Noting your comment in the text that 87% were expressed in the aorta (were the 37 genes?), this is worth commenting on. As you brought forward all the genes at the 121 loci for enrichment testing, it would be interesting to see your results without the 37 genes based on distance only if not expressed in aorta.
Response	Twenty eight of the 37 (75%) of the genes that were prioritized based only on distance are expressed in aorta. We have added this information in the text. Per the reviewer's suggestions we have also now performed a gene ontology enrichment analysis (Table S15) based on the 84 genes that were prioritized by multiple measures (by protein altering variant or consensus) and identify similar GO terms that were discussed in the result section.
Excerpt from manuscript	Most of the prioritized genes (80 of 84 genes prioritized by protein-altering variation or consensus, 28 of 37 genes prioritized by distance only) were expressed in aorta. We performed a sensitivity analysis using 84 genes that were prioritized by protein-altering variation or consensus (Table S15).

11. Gene prioritization

Comment	Section: Gene prioritisation identifies putative causal genes at AAA risk loci. I noted that for the secondary signals you included genes only based on distance why was the same process not followed for selection of candidate genes as the lead signal?
Response	Our emphasis in the current work is on the discovery of most likely causal genes underlying the AAA associated variants rather than the variants themselves. Accordingly, we did not thoroughly explore the secondary signals at the existing loci and did not perform gene prioritization for them as we do not feel this would add significantly in identifying novel AAA related genes. Additionally, the statistical analyses to rigorously undertake this would require new methods development as current fine-mapping tools are shown not to work adequately for meta-analyzed data and therefore outside of the scope of the current paper.

12. TWAS

Comment	Section: Transcriptome-wide association study indicates the direction of gene-expression changes of AAA associated genes You performed a TWAS in aortic tissue leveraging 104 genes these were then explored looking at differential expression in mouse and human tissues. The outcome from this analysis was 23 genes with consistent directions of effect and only 11 of these were prioritised as causal genes from your earlier pipeline. Subsequent work validated the 11 genes. I think you need to include your motivation for the TWAS as a method for prioritising candidate genes as a separate analysis to the pipeline you followed above – why this was done needs to be clearer as others using this tool as a way to prioritise genes and you could have used the results as an indicator. The results differ – with little overlap, validation of genes from TWAS used same datasets as your earlier pipeline. I understand you wish to highlight the most promising candidate genes as best you can with the existing data but some discussion on the motivation for each of your analyses is required and why results are discrepant, now only 11 genes and
--

	these are the ones you focus discussion on. Can you map back the 11 genes as having eQTLs from stage 1?
Response	Our interpretation of this comment is that the reviewer is asking 3 separate but related questions -- A) Why was TWAS reported as a separate experiment in addition to the multimodal gene prioritization exercise? B) Why are a higher number of genes mentioned in TWAS+mouse model compared to 11 prioritized genes that were followed up by qPCR? C) How many of the 11 genes investigated by qPCR had support of eQTLs? Taken in parts: A) Why was TWAS reported as a separate experiment in addition to the multimodal gene prioritization exercise? As TWAS integrates genomic data and results in gene level association it provides information that is complementary to traditional GWAS with respect to gene identification; this is not limited to genes at GWAS significant loci. Additionally, it provides information on direction of gene expression (up or down regulated) that is harder to discern from variant level analyses. Because of the rich nature of the data, we felt that there was value in presenting the TWAS results as their own experiment in addition to using them aid in gene prioritization at GWAS loci. As a note, at the stage 1 of gene prioritization, support of either colocalization or TWAS association in 121 loci was counted as one piece of evidence (see figure 3a for eQTL by colocalization/TWAS). B) Why are a higher number of genes mentioned in TWAS+mouse model compared to 11 prioritized genes that were followed up by qPCR? While we feel that the TWAS, and its joint analysis with mouse transcriptomic data is interesting, we were interested in pursuing only the most promising causal genes in de novo qPCR experiments; the desire to pursue only a limited number of genes was motivated by both statistical (multiple testing) and logistical concerns. For this reason, we took the intersection of the TWAS+mouse analysis and the gene prioritization analyses to select genes for de novo replication. We have modified the text (see below) to make our treatment of this data, and the rationale behind it, clearer.

	C) How many of the 11 genes investigated by qPCR had support of eQTLs? Of the 11 genes investigated by qPCR, 8 demonstrated evidence of formal eQTL colocalization.
Excerpt from manuscript	GWAS often identify non-coding variants that are hypothesized to be associated with disease via alteration of gene expression levels of causal genes. To further explore this possibility, TWAS methods leverage a predictive model of gene expression from a reference panel (eQTLs) to predict gene expression from GWAS summary statistics, followed by a test of association between predicted expression and phenotype²⁸. These associations in 121 genome-wide significant loci were used as an indicator in the gene-prioritization step as described above (Figure 3a). In addition, TWAS also indicates whether up or down-regulation of a gene is associated with the trait (Table S6). To identify consistency of the direction across datasets, we further inquired whether genes identified by TWAS (aortic tissue reference panel²⁹) were differentially expressed in aortic tissue from mouse models of AAA (Table S11). Of 121 prioritized genes, 22 had support of both TWAS and mouse model of AAA. Of 22 genes supported by both, 10 genes were differentially expressed in the same direction during mouse AAA development and TWAS prediction. Additionally, 4 genes displayed mixed direction in the mouse model, i.e., direction matched with TWAS for some experimental conditions. Out of these 14 genes that displayed consistency in direction, 11 (Figure 4a) were prioritized as likely causal genes at GWAS loci by our consensus approach without distance as a tie-breaker as described previously.

Reviewer 3

General comments: The authors have sufficiently addressed all my comments and have amended the manuscript accordingly.

Decision Letter, second revision:

19th May 2023

Dear Dr. Damrauer,

Thank you for submitting your revised manuscript "GWAS meta-analysis deciphers genetic architecture of abdominal aortic aneurysm and highlights PCSK9 as a therapeutic target" (NG-A60516R1). It has now been seen by the original referees and their comments are below. The reviewers find that the paper has improved in revision, and therefore we'll be happy in principle to publish it in Nature Genetics, pending minor revisions to comply with our editorial and formatting guidelines.

Sincerely,
Wei

Wei Li, PhD
Senior Editor
Nature Genetics
New York, NY 10004, USA
www.nature.com/ng

Reviewer #2 (Remarks to the Author):

Many thanks to the authors for responding to my additional queries, I am satisfied with the responses and the changes made to the manuscript.

Final Decision Letter:

22nd Aug 2023

Dear Dr. Damrauer,

I am delighted to say that your manuscript "Genome-wide association meta-analysis identifies risk loci for abdominal aortic aneurysm and highlights PCSK9 as a therapeutic target" has been accepted for publication in an upcoming issue of Nature Genetics.

Your paper will be published online after we receive your corrections and will appear in print in the next available issue. You can find out your date of online publication by contacting the Nature Press Office (press@nature.com) after sending your e-proof corrections. Now is the time to inform your Public Relations or Press Office about your paper, as they might be interested in promoting its publication. This will allow them time to prepare an accurate and satisfactory press release. Include your manuscript tracking number (NG-A60516R2) and the name of the journal, which they will need when they contact our Press Office.

Please note that *Nature Genetics* is a Transformative Journal (TJ). Authors may publish their research with us through the traditional subscription access route or make their paper immediately open access through payment of an article-processing charge (APC). Authors will not be required to make a final decision about access to their article until it has been accepted. [Find out more about Transformative Journals](https://www.springernature.com/gp/open-research/transformative-journals)

Authors may need to take specific actions to achieve [compliance](https://www.springernature.com/gp/open-research/funding/policy-compliance-faqs) with funder and institutional open access mandates. If your research is supported by a funder that requires immediate open access (e.g. according to [Plan S principles](https://www.springernature.com/gp/open-research/plan-s-compliance)) then you should select the gold OA route, and we will direct you to the compliant route where possible. For authors selecting the subscription publication route, the journal's standard licensing terms will need to be accepted, including [self-archiving-and-license-to-publish](https://www.nature.com/nature-portfolio/editorial-policies/self-archiving-and-license-to-publish). Those licensing terms will supersede any other terms that the author or any third party may assert apply to any version of the manuscript.

If you have not already done so, we invite you to upload the step-by-step protocols used in this manuscript to the Protocols Exchange, part of our on-line web resource, natureprotocols.com. If you complete the upload by the time you receive your manuscript proofs, we can insert links in your article that lead directly to the protocol details. Your protocol will be made freely available upon publication of your paper. By participating in natureprotocols.com, you are enabling researchers to more readily reproduce or adapt the methodology you use. [Natureprotocols.com](http://natureprotocols.com) is fully searchable, providing your protocols and paper with increased utility and visibility. Please submit your protocol to <https://protocolexchange.researchsquare.com/>. After entering your [nature.com](http://www.nature.com) username and password you will need to enter your manuscript number (NG-A60516R2). Further information can be found at <https://www.nature.com/nature-portfolio/editorial-policies/reporting-standards#protocols>

Thank you.

Sincerely,
Chiara

Chiara Anania, PhD
Associate Editor
Nature Genetics
<https://orcid.org/0000-0003-1549-4157>